# Identification and Simulation of the Influencing Factors of Private Capital Participation in Urban and Rural Infrastructure Transformation Based on System Dynamics

**Hui Chen** [1,*]**, Yuxuan Zhu** [2]**, Xiaoqing Du** [2]**, Hong Yan** [2] **and Guanghui Fu** [2]

[1] Institute of Finance and Public Management, Anhui University of Finance and Economics, Bengbu 233030, China
[2] School of Economic and Management, Nanjing Tech University, Nanjing 211816, China; zhuyuxuan@njtech.edu.cn (Y.Z.); duxiaoqing@njtech.edu.cn (X.D.); yanhong@njtech.edu.cn (H.Y.); fugh@njtech.edu.cn (G.F.)
[*] Correspondence: chenhui@aufe.edu.cn

**Abstract:** An important force for participation in urban and rural infrastructure transformation (URIT) is private capital (PC), which helps to emphasize the importance of government departments in effectively implementing quality urban development strategies when driving PC in order to participate in URIT in a compliant and efficient manner. This article constructs a system of factors that influence the participation of PC in URIT, which uses the analytical hierarchy process–criteria importance through inter-criteria correlation (AHP-CRITIC) combination method to quantify the comprehensive assignment of influencing factors, in order to analyze the poor effectiveness of PC participation in URIT. On this basis, combined with a logical mechanism analysis of PC participation in URIT, the evolutionary system dynamics model was constructed, and it concerned the correlation between PC's willingness to participate in URIT and PC's willingness to participate in each of the influencing factors. The results of the study show that (1) factors such as "return on project investment", "tax breaks", "level of government finance", "project construction cost", "mechanism for public selection of PC", and "establishment of a special coordinating department" are the most influential factors concerning the willingness of PC to participate in URIT; (2) the "open selection of PC" and the "establishment of a coordinating department" at the level of government behavior, and "tax relief" at the level of the policy system, directly affect the willingness of PC to participate in URIT; and (3) the analysis is based on the model simulation results, combined with stakeholder theory and incentive theory. After analyzing the simulation results, it was observed that increasing the degree of implementation, regarding the "public selection of PC" and "establishment of coordination departments" among the influencing factors related to the government's behavior, significantly enhanced the willingness of PC to participate during the final stage of the simulation. The willingness increased from 2.81 to 3.24 and 3.22, respectively. Furthermore, after doubling the "tax relief" within the policy system, the willingness of PC to participate increased from 2.81 to 3.05 during the final simulation. Finally, this article gives strategic recommendations as to how governments can incentivize PC participation in URIT, which mainly include strengthening the profitability of URIT projects, innovating the PC participation model, improving the completeness of the supporting policies, and strengthening the co-ordination of local policies. The theoretical models and research results presented in this article can provide a reference for government policy makers to encourage PC participation in URIT and provide new ideas for transformation methods concerning PC participation in URIT.

**Keywords:** private capital; urban and rural infrastructure transformation; influencing factors; incentive strategies; system dynamics

## 1. Introduction

URIT has been popular for almost half a century worldwide and, in recent years, large-scale URIT projects have been carried out in many developing countries. Given the rapidly increasing rate of urbanization, competition among the expanding population for limited resources is bound to occur in cities [1]. Since the reform and opening up of China, the development of Chinese cities has mainly occurred in terms of an outward expansion of urban boundaries, and hard constraints on spatial resources have bottlenecked the development of most cities. The traditionally crude, extensive, and incremental expansion mode can no longer meet the needs of today's social development, and the main approach to urban development is gradually moving towards URIT. As a sustainable approach to urban transformation, URIT can improve resource utilization, mitigate urban decline, and optimize urban spatial structure [2]. Urban sprawl is being replaced by URIT as the core problem facing the world's cities as the global urbanization rate gradually exceeds 50% [3]. The endowment of a city's hard infrastructure may not play a decisive role in enhancing competitiveness; however, social infrastructure promotes wellbeing and improves the quality of the population, thereby enhancing the performance of urbanization [4].

URIT improves existing urban areas; thus, it is a sound approach to cope with urban decay and achieve multiple socioeconomic goals [5]. The principal objectives and inner mechanism of URIT have undergone a profound transformation in recent years. Synthesis and integration are concepts embodied by URIT. URIT pays more attention to certain issues such as the development of connotation, upgrading quality, the intensive use of land, and the transformation and upgrading of the industry in a city. Governments have successively put forward strategic requirements such as successfully carrying out urban repair, improved management, and high-quality development in order to introduce the participation of PC in an effective way. Government departments are the leading parties and decision-makers when it comes to URIT-integrated management, and they have been conducting relevant research into PC participation with regard to URIT incentives. In 2023, the Ministry of Housing and Urban–Rural Construction of China issued a document highlighting that urban design should be regarded as an important means of URIT: it should improve the urban design management system; specify the design requirements for buildings, districts, communities, blocks, cities, and villages on different scales; propose the design conditions for the construction and transformation of urban and rural infrastructure plots; and organize the preparation of key project design plans in order to regulate and guide the implementation of URIT projects.

In China, PC participation in URIT is challenging due to its unique market system and social culture [6]. With the rapid development of the external economic environment of cities, and the continuing reform of the government–enterprise co-operation model for URIT in recent years, PC still faces more problems in terms of URIT implementation, which is especially highlighted by the advancement of micro-profitable renewal projects with lower rates of return that are relatively difficult to implement. Some projects, even if successfully implemented under the participation of government or capital forces, increase the risk of urban poverty and social isolation [7]. Therefore, exploring the factors influencing and logically incentivizing PC to participate in URIT, as well as the incentives guiding the constant flow of PC to URIT, is a strategic choice to promote the connotative development of the city. It is also practical in the sense that it promotes the high-quality development of the city, and it comprises the core of the development of URIT, which is optimized by the government.

Currently, the existing methods of PC participation in URIT and urban research can be summarized in terms of three main areas. The first area concerns social networks that are measured by individuals' internal and external social relationships in their residential neighborhoods [8,9]. Second, norms of reciprocity and trust are considered important consequences of urban development [10]. The third domain is community participation, which is measured in terms of both attitudinal and behavioral dimensions [11]. The inevitable trend of URIT is to move from monolithic governmental dominance to the

synergistic participation of multiple actors, making the force of the market the main force of URIT that is necessary to guarantee its sustainable operation. In the context of URIT, research confirms that the establishment of a social environment can be assisted by PC, which generates social norms of cohesion and general trust and facilitates co-operative behavior [12,13]. Thus, PC is crucial in terms of the success of URIT projects.

However, so far, scholars at home and abroad have carried out less research on the specific transformation mode of PC participation in URIT and systematic incentive strategies; especially lacking is an understanding of the channels through which PC participates in URIT, what the factors influencing PC participation are, and how PC can be effectively incentivized to help URIT in order to achieve a win–win situation, along with other core issues. The implementation and efficiency of URIT may be affected, and the exploration of different mechanisms of URIT is challenging under conditions that have not been deeply analyzed in terms of these issues. Therefore, this article presents a study on PC participation in URIT with a focus on addressing the identification of influencing factors and the construction of incentive strategies for PC participation in URIT.

This study aims to address the following research questions:

1.  What are the factors influencing PC participation in URIT? What is the relationship between the influencing factors?
2.  What are the key factors of PC participation in URIT?
3.  How does government behavior affect PC's willingness to participate?
4.  What are the government's proposed countermeasures to incentivize PC participation in URIT?

This paper addresses the study of PC participation in URIT, broadens the research content of the new participation architecture model of PC in sustainable urban and rural development, provides effective theoretical support for further improving the systematic countermeasures of and suggestions for URIT, and provides a dynamic simulation evolutionary path analysis paradigm for the quantitative assessment of the willingness of PC participation in URIT. In practice, this will help encourage the government to better utilize the power of PC for URIT and achieve a win–win situation between the government and PC.

## 2. Literature Review

### 2.1. URIT Research

Urbanization is one of the most important processes determining global environmental change in the current era. The multiple ways for sustainability outcomes to be generated by the interaction of rural and urban processes need to be understood to advance global sustainable development [14]. The World Bank classifies infrastructure as both economic and social infrastructure. According to the definition by the Department of Infrastructure and Planning, Queensland, Australia [15] (2009), urban and rural infrastructure (URI) refers to "urban and rural facilities, services and networks that help individuals, families, groups and communities to meet their social needs and maximize their development potential and enhance well-being".

The theory of land-use transformation holds that urban and rural transformation depends to a large extent on the development of infrastructure, which can promote economic growth [16], and the green infrastructure environment has a positive impact on people's cognitive function [17]. Research on rural–urban interactions mostly assumes urbanization to be a main driver of global sustainability, and infrastructure to be a main mechanism that enables urbanization [18,19]. The main underlying intentions and motivations of PC investment in URIs are market expansion and profit maximization [20], as well as a return on investments being paid through payments or subsidies from government and user fees [21].

Bramley and Power [22] argued that urban sustainability is relevant to urban spaces, which is reflected in people's interests and community development. As asserted by Nzimande [23], URIT should be used to maintain social stability through the participation

of both the public and stakeholders. Babatunde et al. [24] found that public–private partnerships (PPPs) can improve the sustainability of public services and the efficiency and availability of resources for several social infrastructure projects. Liang Ma et al. [25] constructed a social infrastructure knowledge system based on the PPP literature from the perspective of knowledge systems, application status, and prospects, and hold that, although the PPP results differ among different topics, they still have valuable application potential in terms of the provision of social infrastructure. S. Baron et al. [26] proposed a joint interdisciplinary research project to implement the long-term transformation of existing water supply, wastewater, and energy infrastructure and to provide the basis for ideas and strategies for visualizing the cross-sectoral optimization of facility transformation. Victor H et al. [27] shifted the concept of the urban–rural relationship from the process of "urban" to "rural" and practiced a locally driven approach of urban–rural sustainable development. It is helpful to inspire a new generation of infrastructure that perceives nature as an essential element of urban sustainable development. Ouyang et al. [28] conducted a hedonic analysis by applying Bayesian model averaging to data from the China Labor Force Dynamic Survey (CLDS) to investigate the impact of social infrastructure capitalization on the value of urban and rural housing in China, and revealed the equal importance of social infrastructure for the value of housing. Decades of study on URIT at home and abroad has found that the continuous improvement of the theoretical system provides guidance and support for its practical operation.

In summary, research on URIT from domestic and foreign scholars has developed from conceptual definitions and methodological exploration to different areas of application, and has achieved fruitful results. Understanding that rural and urban places, practices, and processes are mutually constitutive can provide pathways for a new generation of infrastructural transformations that facilitate the transition of rural–urban systems toward sustainable development [29]. The entry point of this paper is PC's participation in URIT, with an analysis of the evolutionary trend of each influencing factor based on a system dynamics model, to provide theoretical support for formulating government incentive policies, and at the same time to provide new perspectives for research on URIT.

*2.2. Study of PC Participation in URIT*

In the 1980s, PC theory began to emerge, and the concept was widely and rapidly applied in the research fields of economics, sociology, management, and political science by foreign scholars. "PC" gradually became an important element of related research on URIT, urban planning, and government policy. The document *Our Towns and Cities: The Future Delivering and Urban Renaissance* was published in 2000 by the UK Department of the Environment, Transport and the Regions, which emphasized the importance of community participation from PC in URIT. Masoumeh Hafiz Rezazadeh [30] found that strengthening urban PC can facilitate the process of achieving the sustainable development of urban areas through the development of urban tourism. Rutheiser C. [31] examined the measures undertaken by public agencies and private enterprises in the city of Atlanta, USA, to establish commercial centers and residential areas to solve the problems of inner city decline and gentrification after the 1996 Olympics. They found that the focus of these measures was only on the transformation of the physical dimension and ignored the fundamental problems of urban decline caused by political, social, and economic factors. Turcu [32] explored the perspective of residents and the government, believing that the participation of external support and PC are useful for the sustainable renewal of communities. Gerhard Hatz [33] examined "gentle URIT", featuring public–private partnerships taken by the government of Vienna. By testing the social sustainability of the subsidized URIT model in Vienna using the indicators of apartment rents, apartment quality, and tenants' household incomes, he found that the generation of the phenomenon of gentrification is inevitable. Margarita Greene [34] argued that differentiated incentives for realtors to participate in URIT should be implemented according to the characteristics of the society and the city in each region. Banerjee [35] studied the economic and legal

system and investment effects in a number of countries, finding that countries with a better economic system can induce more infrastructure projects in which PC can invest on the premise of the same factors, as also verified by Panayides et al.'s study [36].

*2.3. Private Capital Participation Model*

1.  Community Development Corporation (CDC) Model

In the 1940s, the old demolition-style city renovation movement dominated by PC was carried out in the United States and included the "transformation of shantytowns" in the early days of the country. However, this large-scale demolition and construction model incurred social dissent very quickly, so the government focused on implementing the URIT model to enhance the welfare of the residents through community action, and the establishment of Community Action Agencies (CAAs) to enhance the amenities of the community by means of government funding. However, the expected results were poor because of the inefficiency and low capacity of the government. Therefore, CAAs began to take over community responsibility instead of the government by means of tapping community resources to enhance the economy, employment, and housing supply, and blending the power of PC and the power of the government to realize the renaissance of the urban community [37].

At the same time as adopting the market operation mechanism, CDCs are also involved in the grassroots management of governments. The operating mechanism of CDCs consists of four aspects: firstly, the concession—the company explains the intended structural organization and provides a detailed plan of community development operations, which is then approved by the government [38]. Secondly, in the governance structure of the modernized enterprise, in terms of the management structure setup and personnel arrangement, the board members serve as technicians of the URIT-related professions and invite external financing institutions to guide and supervise the operations [39]. At the same time, the financial situation and operational planning are monitored by the public. Thirdly, in the operation of community assets, different from real estate development enterprises' pursuit of excessive profits, PC is used mainly through the development of basic security housing and community business operations, childcare, health care, and other service industries to generate income and to preserve and enhance the value of the assets through operational diversification. Fourthly, due to the convenience of the policy and financing provided by the government, tax exemptions will be provided to the PC in order to maintain a balance of revenues and expenditures and to be financed with assistance from community development grants and the Ford Foundation.

Although the operation of CDCs has been greatly affected by the level of urban economic development, there are still surviving CDCs that are responsible for the construction and operation of US communities under the support of state and local funds, which suggests that some cities' grassroots affairs can be operated by the marketplace and that the efficiency of their operations can be enhanced under the appropriate incentives provided by the government when safeguarded by their supervision.

2.  Enterprise Zone (EZ) and Business Improvement District (BID) models

In the 1970s, in response to the widespread industrial and commercial decline in urban centers, two special zones consisting of enterprise zones and business improvement districts were designated as areas of decline by the government and were assigned special district renewal mechanisms and economic incentives to revitalize the districts.

The enterprise zone mechanism originated in the United Kingdom. It was established by the UK government, with the initial site selection focused on areas of significant economic decline and the spatial decay of inner cities that resulted from the retreat of manufacturing. The main premise of the enterprise zone mechanism is to increase tax policy concessions and reduce government control to stimulate business vitality, leading to a ten-year period of incentivized fiscal and planning facilitation policies within the enterprise zone, such as simplified construction planning procedures, the expansion of tax credits,

and exemption from business tax, thereby promoting regional economic development [40]. Prior to the establishment of business improvement districts, the proposed services, expected performance, management bodies, and organizational structure of the area should be clarified for local businesses and business operations. An application should then be submitted for the establishment of business improvement districts, and a vote taken by business entities in the area on whether or not to set one up; these are mainly established in the retail industry, residential concentration areas, or industrial areas. Its operation mechanism is different to that of enterprise zones, which are mainly invested in by professional management organizations who collect special tax funds from enterprises to maintain public affairs activities including social security and the improvement of the operating environment of commercial spaces [41]. Both of these regional mechanisms are delivered by the government and the franchise management agency responsible for the area. Such an agency, on the one hand, implements the specific policies of the region and is responsible for ascertaining the policy preferences of owners in the area; on the other hand, the agency promotes regional economic development through ways of improving infrastructure and promoting community services, commercial marketing activities, etc. [42].

3.    Municipal Land Organizing (MLO) Model

With respect to the relevant policies and practices of incentivizing PC participation in URIT, foreign countries have explored the municipal land consolidation model, which has also been well implemented in Taiwan, China. Since the end of the 19th century, municipal land organization, as a tool of urban planning, was gradually promoted and widely used in Germany. The enactment of the Land Consolidation Act by Germany in 1953 effectively promoted the development of German cities [43]. From the 20th century, many countries began to use tools of municipal land consolidation to carry out adjustments to optimize the planning of land use and urban development. For example, the Urban Development Policy of the United Kingdom encouraged co-operation between the government and local communities in the 1970s to help communities to carry out more development on their own.

Although the concept and connotations of municipal land consolidation vary in different regions, comprehensively speaking, municipal land consolidation is the rational organization of the development and utilization of urban land on the basis of existing land, carrying out the recombination of spatial configuration, internal elements, land tenure, and revenue of urban land, thus promoting orderly and intensive land use in the city and improving the economic carrying capacity and output yield of the land. This is an important technology for optimizing urban land use; related supporting policies include land concessions and volume rate transfer to incentivize PC to assist the government in implementing the URIT. For example, the "plot ratio bank" in the United States treats plot ratio as a special kind of real estate, with storage purchased by the executive body of the government or authorized by the government, and the allocation or transfer being carried out according to the demand of urban development, taking into account the interests of the developer who purchased the property rights, the land title owner, the neighborhood, the public, and the government during implementation [44]. Japan has also introduced a similar system in which the utilization rights of the residual floor area ratio are traded, allowing landowners and PC subjects, under the premise of compliance with the requirements, to formulate regional development plans according to their own needs, taking into account the public benefits of the city through further relaxation of the restrictions of floor area ratios to incentivize the market mainstay in the capital efficiency of URIT [45].

## 3. Methodology

### 3.1. Research Design

The transformation of infrastructure in urban and rural stock cannot be sustained by the limited financial power of the government alone. In areas that are in urgent need of transformation, but where the rate of return on transformation is low and implementation is relatively difficult, it is often necessary to guide the flow of market capital through incentives. A core issue that urgently needs to be explored and resolved is how to make

flexible use of incentive strategies in order to fully utilize market resources, adjust the relationship between land property rights, balance the interests of all parties, and improve the efficiency of urban resource allocation.

Therefore, from the perspective of efficient government regulation, in view of the current status quo of the low motivation of PC participation in URIT, we identify the influencing factors of PC participation in URIT and determine the influencing weights, and establish the system dynamics evolution architecture of PC participation in URIT. Through software simulation analysis, this study observes the change in the development trend of PC's willingness to participate in URIT under the role of various influencing factors, and accordingly puts forward targeted countermeasures and suggestions for governments to incentivize PC participation in URIT. In this way, governments can solve practical problems such as the insufficient consideration of incentive strategies and the unclear participation mechanisms that exist when encouraging PC to participation in URIT and developing market-oriented transformation, thus promoting the process of government-led and market-operated URIT.

In this paper, a literature analysis is used to summarize the relevant literature concerning URIT, and on this basis, the research results regarding the influencing factors of PC participation in URIT from various studies are statistically summarized. Then, through the combination of AHP and CRITIC analysis, subjective and objective judgments are made on the weights of the influencing factors of PC participation in URIT, and the empirical knowledge and abstract judgment of experts are visualized for data processing. Finally, on the basis of system dynamics, we construct a model of factors influencing the change of PC's willingness to participate in URIT, so as to more intuitively describe the process of the dynamics of this change and provide a model basis for the construction of the government's strategy for encouraging PC's participation in URIT. The technical route of this study is shown in Figure 1.

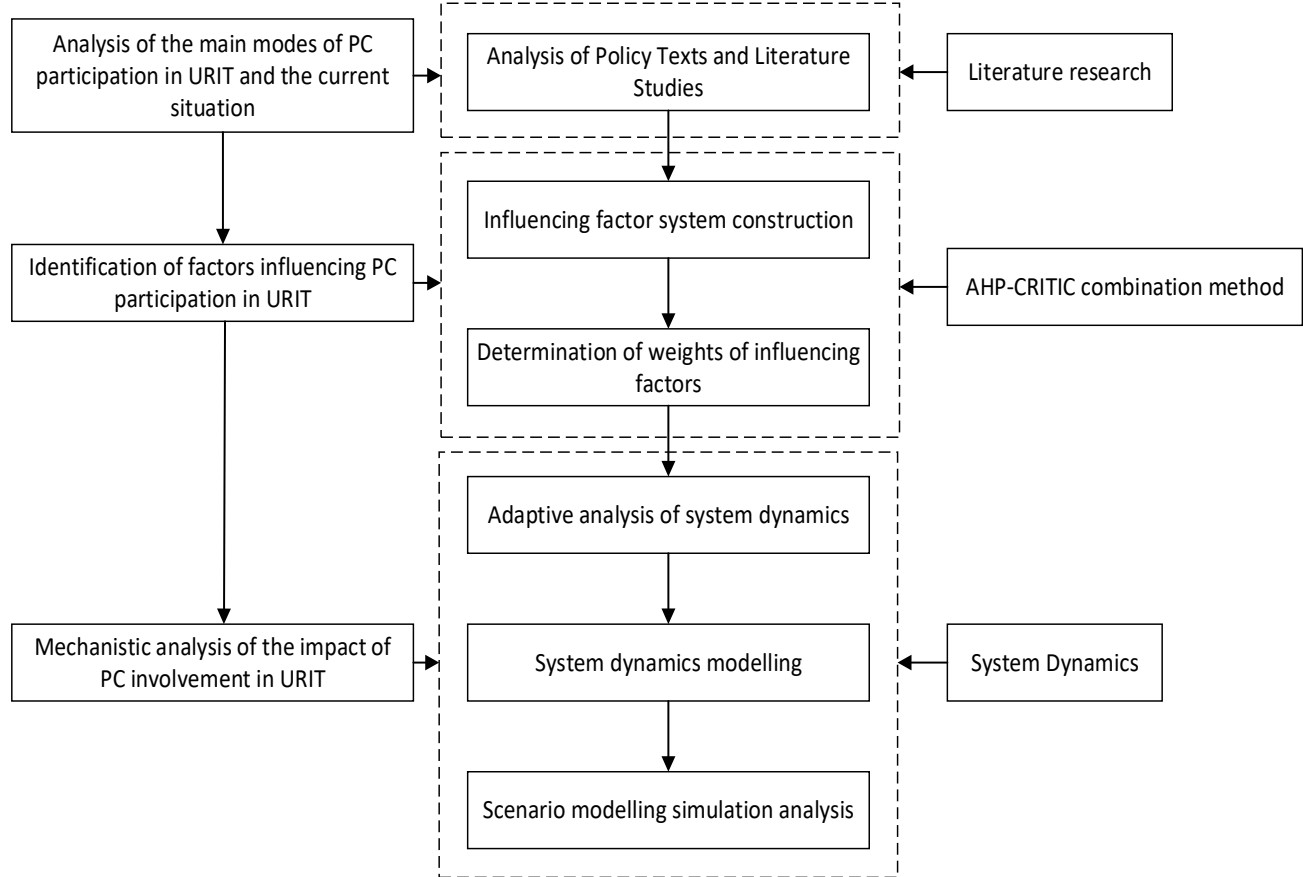

**Figure 1.** Technology roadmap.

### 3.2. Theoretical Framework

System dynamics is mainly used to study the behavioral patterns and characteristics of a system. It relies on internal information feedback control within the system and utilizes computer simulation technology to investigate the dynamic behavior of system development. Decision support can be provided by analyzing the changing trends of the complex system under different parameters or strategic factors [46]. In recent years, system dynamics models have been widely applied to the study of trends in the development of abstract concepts, such as data security impact factor analysis [47], the evolution of online public opinion dynamics, and residents' willingness to participate in waste separation. Many scholars apply system dynamics to the field of URIT to study the mechanism of spatial production in cultural and historical districts, URIT policy evolution, implementation effectiveness, and URIT decision system simulations. The research model for this study is presented in Figure 2.

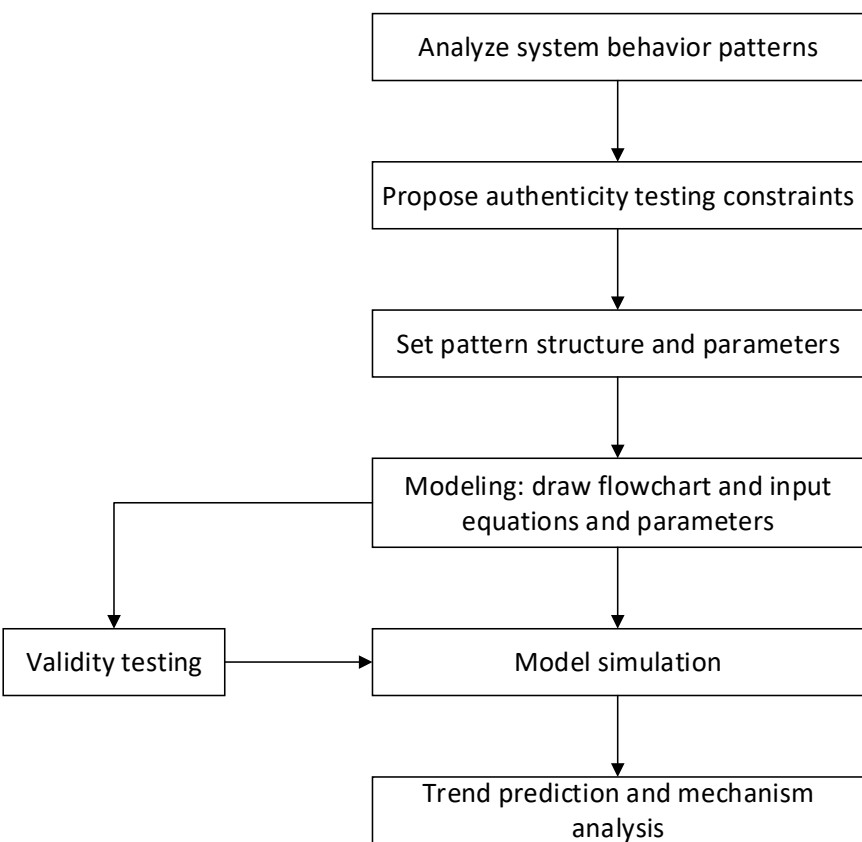

**Figure 2.** Research framework of system dynamics.

### 3.3. Influencing Factor Identification

3.3.1. Analysis of Influencing Factors

To realize the completeness and wholeness of URIT strategy construction, the systematic integration of the influencing factors of PC participation in URIT is necessary. Based on the research on policy analysis and literature combing, combined with the distribution of participating subjects of URIT, this study synthesized and condensed the composition of influencing factors, and strengthened the intrinsic correlations and the role of targeting influencing factors from the perspective of government and enterprise co-ordination and co-operation. This was achieved using four levels of integration and combing, including the policy making of government and enterprise co-ordination and co-operation, government behavior, project situation, and enterprise management, merging duplicate factors and deleting parts of the factors that are less mentioned in the relevant literature and policy texts, to arrive at the final identification of the influencing factors (as shown in Table 1).

**Table 1.** Factors influencing PC participation in URIT.

| Serial Number | Dimension | Influencing Factors | Explanation |
|---|---|---|---|
| 1 | | Allow the transformation of land use nature and function | To safeguard public interests and security, the nature and purpose of land use can be adjusted according to procedures |
| 2 | | Credit support from financial institutions | Organize and co-ordinate financial institutions to increase financial support for all kinds of renovation work |
| 3 | | Interest subsidies | Financial subsidies and other revenue agreement rules disclosed in advance |
| 4 | | Volume ratio index regulation | If it is difficult to achieve economic balance due to the need for protection, then volume ratio transfer is allowed to meet the policy requirements |
| 5 | Policy system | Flexible delineation of land boundaries | Land boundaries can be delineated flexibly depending on circumstances |
| 6 | | Tax reduction and exemption | Projects that meet certain conditions shall enjoy preferential tax policies in accordance with the law |
| 7 | | Simplification of approval process | Combined with the reform of the examination and approval system, streamline the approval items and procedures of URIT projects |
| 8 | | Targeted policies for different types of projects | Update policies for different types of renewal, such as renovation of old factories and urban ecological repair |
| 9 | | Clear delineation of property rights | Ownership of land and housing property rights before renewal, and ownership of property rights such as new construction area |
| 10 | | Policy propaganda and interpretation | The government propagates and interprets policies, especially relevant preferential policies, by means of news platforms |
| 11 | | Clear management boundary between government and enterprises | Clearly delineate the responsibilities of the government and enterprises involved in the project |
| 12 | Government action | Establish a special co-ordination department | Ensure that URIT work is co-ordinated by a certain department or unit |
| 13 | | The spirit of government performance | Reflect the credibility of the government |
| 14 | | The mechanism of publicly selecting PC | Clarify the qualification requirements for selecting PC and ensure that the process is open and transparent |
| 15 | | Government financial level | Reflects the government's ability to pay |
| 16 | | Project construction cycle | Project construction cycle |
| 17 | | The introduction and operation of post-project industry | Some projects need to balance income and expenditure through follow-up operations |
| 18 | Project Status | Relative project cost | The extent of project construction cost relative to the total project investment |
| 19 | | Return on project investment | The return that enterprises can obtain from participating in URIT projects |
| 20 | | The risk of project change | There may be risks throughout the project |
| 21 | | Level of resident collaboration | Lack of co-operation of residents with the project, such as demolition and relocation, may impede the progress of the project |
| 22 | | Enterprise technology level | Enterprise's construction technology in historical building conservation and other aspects |
| 23 | | Enterprise financial situation | The enterprise itself has strong capital to take on risks and changes |
| 24 | Entrepreneurial capabilities | Enterprise financing ability | Enterprises can obtain credit financing to ensure sufficient project funds |
| 25 | | Risk identification and control ability | The enterprise has the ability to identify and effectively prevent risks that may occur during the project construction or operation in advance |
| 26 | | Corporate social responsibility | Enterprises believe that participating in URIT projects is a kind of |
| 27 | | Government–enterprise relationship | mutual trust between the government and enterprises is conducive to co-operation |

Note: The policy and institutional level refers to the relevant supporting policies and institutional processes of URIT formulated by the government, including policy systems such as land policy, planning policy, financial policy, and tax policy, as well as project approval processes and property rights division systems. To unify the index hierarchy of influencing factors, the relevant factors are directly listed instead of dividing them according to policy level and system level.

### 3.3.2. Determination of the Weight of Influencing Factors

Spindle codes are also called quadratic codes. Spindle coding is a process of quadratic analysis. The initial concepts obtained from the open coding were integrated, and 24 spindle codes were extracted from the original open coding.

In this study, using a combination of subjective and objective methods, the AHP-CRITIC combination method was used to determine the weights of the above-mentioned influencing factors. To combine subjective and hangable weights effectively, a degree of difference equation was established based on the Euclidean distance function:

$$d(\omega_{i1}, \omega_{i2}) = \sqrt{\frac{1}{2}\sum_{i=1}^{n}(\omega_{i1} - \omega_{i2})^2} \tag{1}$$

$$d(\omega_{i1} - \omega_{i2})^2 = (\alpha - \beta)^2 \tag{2}$$

$$\alpha + \beta = 1 \tag{3}$$

$$\omega_i = \alpha\omega_{i1} + \beta\omega_{i2} \tag{4}$$

where the subjective weight is $\omega_{i1}$, the objective weight is $\omega_{i2}$, the distribution coefficients of subjective and hangable weights are $\alpha$ and $\beta$, respectively, and the weights sought are $\omega_i$. From the subjective and objective weights that were obtained by the previous calculations, it can be obtained that $\alpha = 0.406$, $\beta = 0.594$, and the combined weights of the AHP-CRITIC can also be obtained, as shown in Table 2.

**Table 2.** AHP-CRITIC combination weight table.

| Serial Number | Dimension | Influencing Factors | AHP Subjective Weight | CRITIC Objective Weight | Combination Weight |
|---|---|---|---|---|---|
| 1 | | Allow the transformation of land use nature and function | 0.032 | 0.036 | 0.034 |
| 2 | | Credit support from financial institutions | 0.033 | 0.043 | 0.039 |
| 3 | | Interest subsidies | 0.038 | 0.038 | 0.038 |
| 4 | | Volume ratio index regulation | 0.033 | 0.042 | 0.038 |
| 5 | Policy system | Flexible delineation of land boundaries | 0.022 | 0.036 | 0.030 |
| 6 | | Tax reduction and exemption | 0.049 | 0.041 | 0.044 |
| 7 | | Simplification of approval process | 0.031 | 0.034 | 0.033 |
| 8 | | Targeted policies for different types of projects | 0.026 | 0.041 | 0.035 |
| 9 | | Clear delineation of property rights | 0.034 | 0.037 | 0.036 |
| 10 | | Policy propaganda and interpretation | 0.022 | 0.034 | 0.029 |
| 11 | | Clear responsibilities between the government and enterprises | 0.038 | 0.029 | 0.033 |
| 12 | Government action | Establishment of special co-ordination department | 0.042 | 0.036 | 0.038 |
| 13 | | The level of government performance | 0.024 | 0.033 | 0.029 |
| 14 | | The mechanism of publicly selecting PC | 0.045 | 0.038 | 0.041 |
| 15 | | Government financial level | 0.044 | 0.040 | 0.044 |
| 16 | | Project construction cycle | 0.028 | 0.033 | 0.031 |
| 17 | | The introduction and operation of post-project industry | 0.044 | 0.042 | 0.043 |
| 18 | Project status | Project construction costs | 0.053 | 0.036 | 0.043 |
| 19 | | Return on project investment | 0.073 | 0.046 | 0.057 |
| 20 | | The risk of project change | 0.040 | 0.037 | 0.038 |
| 21 | | Level of resident collaboration | 0.040 | 0.036 | 0.038 |
| 22 | | Enterprise technology level | 0.024 | 0.033 | 0.029 |
| 23 | | Enterprise financial situation | 0.041 | 0.041 | 0.042 |
| 24 | Entrepreneurial capabilities | Enterprise financing ability | 0.044 | 0.041 | 0.042 |
| 25 | | Risk identification and control ability | 0.036 | 0.035 | 0.037 |
| 26 | | Corporate social responsibility | 0.017 | 0.030 | 0.026 |
| 27 | | Government–enterprise relationship | 0.031 | 0.035 | 0.035 |

The main factors influencing PC participation in URIT include "return on project investment", "tax breaks", "level of government finance", "project construction costs", "late introduction and operation of the project", "corporate financing capacity", "corporate financial status", "mechanism of public selection of PC", "credit support from financial institutions", and "establishment of a special coordinating department". The factors that have a relatively minor impact on PC participation in URIT include "corporate technical level", "government spirit of compliance", "corporate social responsibility", and "policy publicity", among others.

Factors that affect the participation of enterprises in URIT are mainly "enterprise financing ability" and "enterprise financial status", from the perspective of the enterprise's own ability. The more influential factors for PC are "return on project investment" and "project construction cost", so it can be seen that a very important limiting factor for PC is the burden of capital, because URIT projects are basically a high-capital investment and the return on investment is unstable.

The influence weights of "government financial level" and "mechanism of public selection of PC" are the highest in terms of government behavior. Therefore, the government's financial capability is more valued by PC. Some of the advantages of PC in fund management, planning and design, and project construction and operation cannot be effectively utilized due to the existence of unfair market competition, thus hindering the participation of PC in the construction of URIT.

Among them, from the perspective of the policy system, the most important influencing factors are "tax relief" and "credit support from financial institutions". If the financing channels for PC can be opened up and PC investment costs can be reduced by reducing taxes to a certain extent, this will play a certain role in incentivizing PC.

*3.4. System Dynamics Equation Construction*

3.4.1. System Dynamics Model

System dynamics is a discipline that has multi-feedback and multi-variable problems, and is used in studies dealing with social, economic, and environmental highly nonlinear, high-level data and integrated studies of large-scale systems at macro and micro levels [48]. Socioeconomic issues, such as the urban development problems of PC participation in urban and rural infrastructure transformation and construction problems, are in line with the object of this study, namely, system dynamics. Problems with the participation of PC in URIT are associated with multiple feedback systems involving multiple stakeholder subjects. The influencing factors exhibit multi-level and nonlinear characteristics, making it inappropriate to use linear equations for directly calculating PC's willingness to participate. System dynamics, however, aligns with the processing and analysis of time-varying data and the study of large-scale system problem characteristics.

System dynamics is a methodology that primarily uses qualitative analysis, with quantitative analysis taking a supporting role, and the two complement each other to gradually deepen problem solving [49]. It is a research methodology that involves systematic thinking, analysis, synthesis, and reasoning. In the previous part of this paper, the factors affecting PC participation in URIT were qualitatively analyzed. In this section, it is necessary to analyze the effect of each factor on the willingness of PC to participate in a more systematic and profound way, and to supplement this with a quantitative analysis.

System dynamics incorporates dynamics and feedback, allowing time to be represented as a co-ordinate and acknowledging the presence of variable feedback loops within the system. The factors influencing PC participation in URIT, such as government finances and the effectiveness of targeted policies, will change over time. There exists a feedback loop between the willingness to participate in PC and the government's management level and policy system. Therefore, this dynamic relationship aligns with the issue of PC participation in URIT.

The establishment of realistic and standardized mathematical logic expressions, based on the data of relevant variables, is a unique feature of system dynamics theory for studying

and solving problems. Even though model-assisted equations may contain elements of semi-qualitative, semi-quantitative, and qualitative descriptions, the variables can still be categorized based on the composition of the system's basic structure. This allows for a clearer establishment of assumptions for policy experiments and the analysis of the existence of the problem. System dynamics has been extensively applied to synthesize and analyze a large number of studies on influencing factors. This section will establish system dynamics equations based on the foundation of previous studies.

System dynamics analyzes the system's operation mechanism and its effects through a functional simulation process to obtain information about the future development of the system. Based on this information, it seeks pathways to solve the problem. The factors influencing the participation of PC in URIT are complex and long-term. Analyzing the specific guiding role of government actions and policy regimes for PC can be achieved using system dynamics. It can also generate model curves to predict future development trends and provide solutions for formulating incentive policies.

In addition, the willingness of PC to participate in URIT is an abstract variable. Many individual influencing factors are difficult to represent with specific data that can be collected. Therefore, system dynamics can be utilized to simulate and analyze these factors even in the absence of highly accurate data, while still effectively predicting the pattern of the system's behavior.

In conclusion, it is feasible to study the factors influencing PC participation in URIT using system dynamics. System dynamics is applied to the study of the factors influencing the participation of PC in URIT. It is possible to establish system dynamics equations by incorporating various factors of governmental behavior and policy regimes. The system's feedback mechanism is utilized to observe the trend of change in the willingness of PC to participate after assigning values to the influencing factors. By controlling different variables and conducting a scenario analysis through software simulation, the willingness intuitively demonstrates the degree of response to changes in different measures, enabling efficient analysis.

### 3.4.2. System Assumptions

The influencing factor system of PC participation in URIT refers to the role and effect of various dimensions of influencing factors. These dimensions mainly include governmental behavior, the policy system, the project's situation, and enterprise capacity; they influence the willingness of PC to participate in URIT.

In fact, the envisioned approach in system dynamics is to define the boundaries of the model and categorize the relevant parts of the research problem into the system. These parts are isolated from the rest of the system environment, and the objects are extracted from a larger scope of research, creating a simplified closed system. In this paper, the influencing factors that affect the participation of PC in URIT are considered the boundaries of the system. Section 3 identifies the main influencing factors as key variables in the model.

To ensure the feasibility of studying the factors influencing PC participation in URIT, it is necessary to formulate certain conditional assumptions. These assumptions help establish a specific environment for studying the dynamic mechanisms of the system's actions and assist in constructing a system dynamics model.

**Hypothesis 1.** *The system of factors influencing PC participation in URIT is a continuous process, and the willingness of PC to participate is influenced by these factors over time.*

**Hypothesis 2.** *The willingness of PC to participate in URIT is primarily influenced by factors within the system boundary, namely, the policy system, government behavior, project situation, and corporate capacity. The impact of other factors is considered to be very weak and negligible.*

**Hypothesis 3.** *The level of government finances has increased each year over time, and this trend remains consistent throughout the simulated time period.*

**Hypothesis 4.** *The government gradually improves the methods and treatment of detailed regulations for the renovation of different types of urban and rural infrastructure, which change over time.*

**Hypothesis 5.** *With the government interpreting the policy of URIT through media publicity and other forms of creating a social atmosphere, the general public will raise their level of awareness regarding URIT. This, in turn, can increase their understanding and co-operation to a certain extent in relation to URIT activities.*

**Hypothesis 6.** *As the project construction cycle lengthens, the associated change in the risk of the project increases accordingly. However, if stakeholders, such as the original property owners, actively co-operate in the project construction, the occurrence of change risks can be reduced to some extent.*

3.4.3. Create Flow Charts

According to the analysis in Section 3, the factors influencing the participation of PC in URIT mainly stem from four aspects: government behavior, the policy system, project situation, and enterprise capacity. The causal relationships among these influencing factors are depicted in Figure 3.

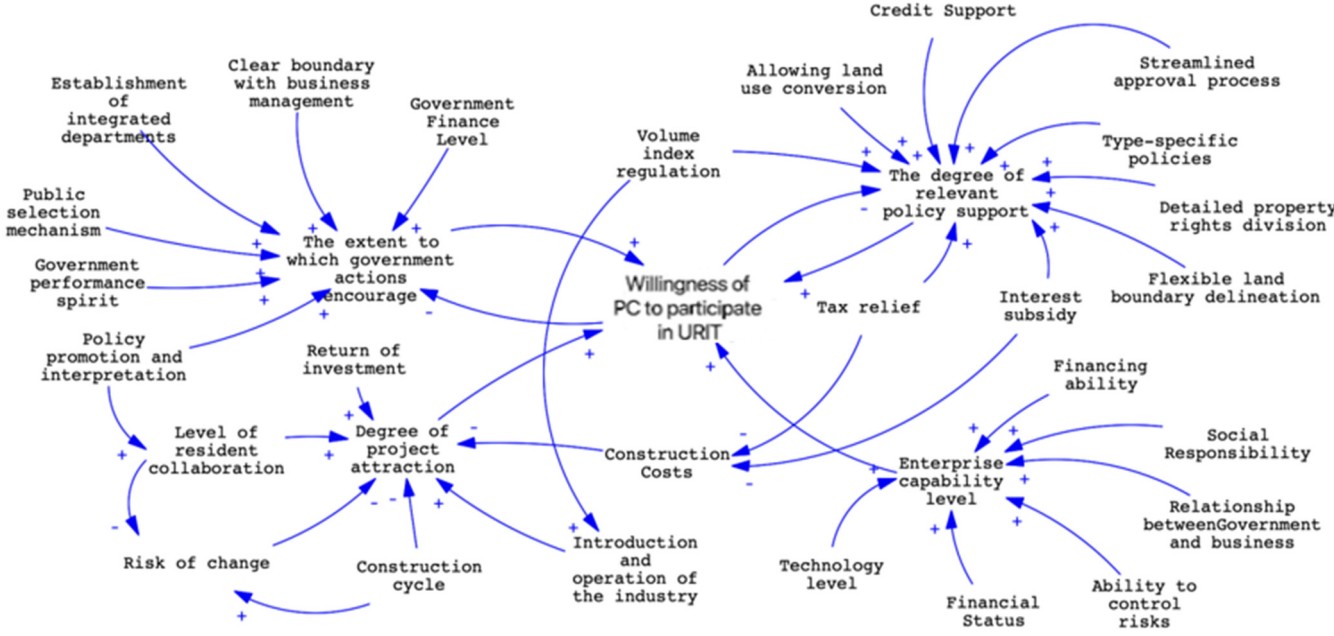

**Figure 3.** System causality diagram of factors influencing PC participation in URIT.

In the system of factors influencing PC participation in URIT, the willingness of PC to participate is influenced by four main dimensions: the degree of governmental co-ordination management, the degree of support from relevant policies, the degree of project attraction, and the level of enterprise capacity. These dimensions, in turn, are each influenced by other factors. There are two main loops in the system:

① The degree of government co-ordination and management ↑—the willingness of PC to participate in URIT ↑—the degree of government co-ordination and management ↓.

② The degree of policy and institutional support ↑—the willingness of PC to participate in URIT ↑—the degree of policy and institutional support ↓.

The willingness of PC to participate in URIT will be strengthened accordingly as the degree of support from the policy system improves. However, once the willingness of PC reaches a certain level, the government may no longer need to provide incentives. This can

lead to a relaxation in the degree of support from the policy for PC, as well as the degree of the government's co-ordinated management.

Based on the causality diagram of the factors influencing PC to participate in URIT, a flow diagram of the system was constructed, as shown in Figure 4.

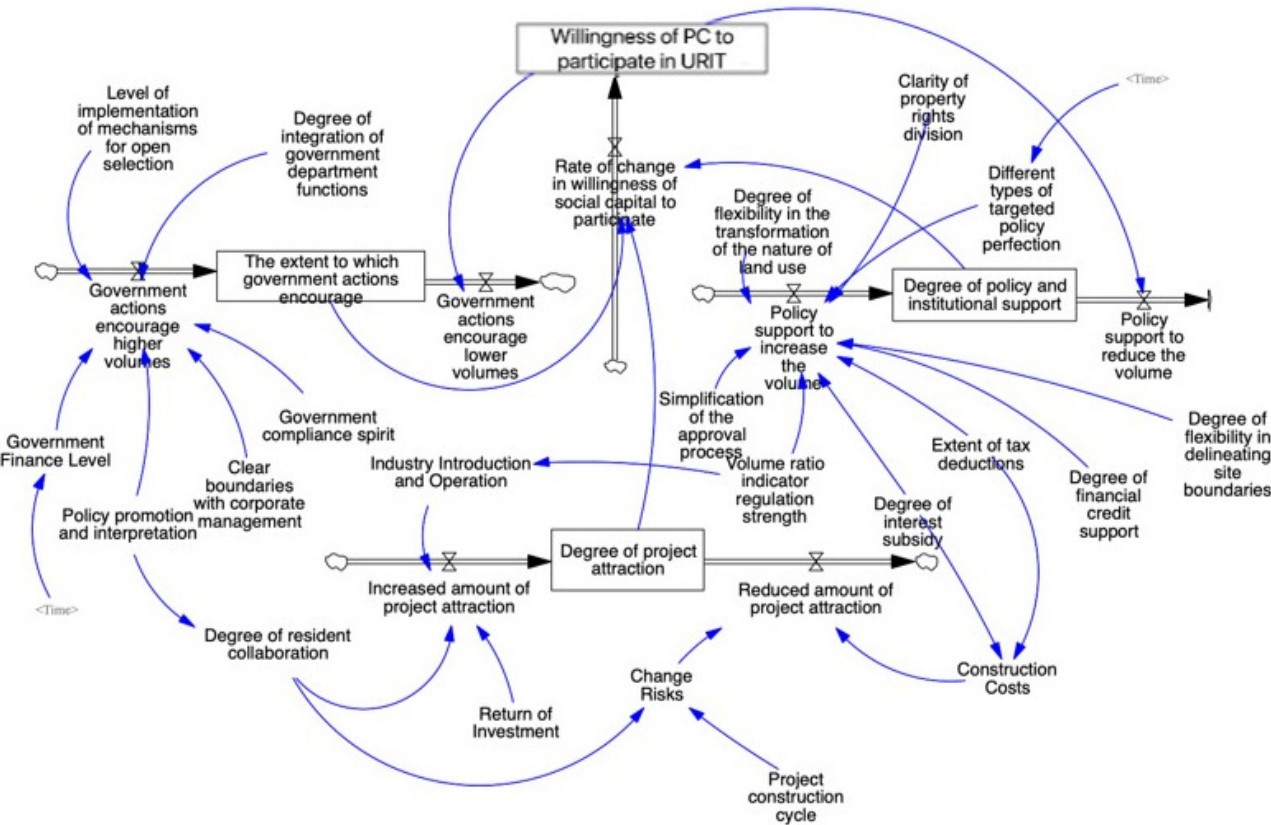

**Figure 4.** System flow diagram of factors influencing PC participation in URIT.

Among the four main influencing factors, namely, policy regime, government behavior, project situation, and enterprise capacity, it is important to note that, although enterprise capacity also plays a significant role in influencing the willingness to participate in URIT projects, it is difficult to directly control or influence the level of enterprise capacity through government behavior and policy regimes related to URIT. Therefore, from the government's perspective, it is practically impossible to stimulate the participation of PC in URIT solely by focusing on improving enterprise capacity. In addition, the level of individual firms' capabilities varies, and it is difficult to take values for influencing factors. Therefore, in the flow chart of establishing the system dynamics model, the main focus is on the influence of the three primary factors—the policy system, government behavior, and project situation—on the willingness of PC to participate in URIT, as well as the process and mode of change. The ability of enterprises is not discussed in this context.

3.4.4. System Dynamics Equation Construction

In this study, the system dynamics model of factors affecting the willingness of PC to participate in URIT includes four state variables, seven rate variables, two auxiliary variables, and eleven constants.

1.   System variable equation

Based on the basic principles of system dynamics and parametric equation models, the definitions of the variational formulas for the influence factors in the system of PC participation in URIT are given in Table 3.

**Table 3.** Definition of system variable equation.

| Nature | Variable Name | Equation Expression |
|---|---|---|
| State variables | The willingness of PC to participate in URIT | INTEG (Change rate of PC participation intention, 1) |
| | The degree of overall government management | INTEG (Amount of increase in policy and system support–Amount of decrease in policy and system support degree, 1) |
| | Degree of policy and system support | INTEG (Amount of increase in policy and system support–Amount of decrease in policy and system support degree, 1) |
| | Project attraction degree | INTEG (Amount of increased project attractiveness–Amount of decreased project attractiveness, 1) |
| Rate variable | Changes in willingness to participate in PC | Degree of government overall management × weight coefficient + degree of relevant policy support × weight coefficient + degree of project attraction × weight coefficient |
| | The amount of improvement in the degree of overall government management | Government financial level × weight coefficient + establishment of planning departments × weight coefficient + open selection mechanism × weight coefficient + government performance spirit × weight coefficient + clear boundary with enterprise management × weight coefficient + policy publicity and interpretation × weight coefficient |
| | The degree of overall government management decreased by a large amount | Willingness of PC to participate in URIT × weight coefficient |
| | Increase in the degree of policy and institutional support | Simplified approval process × weight coefficient + flexible transformation of land use nature × weight coefficient + clear division of property rights × weight coefficient + different types of policies × weight coefficient + regulation of floor area ratio index × weight coefficient + interest subsidy × weight coefficient + tax relief × weight coefficient + financial credit support × weight coefficient + flexible demarcation of land use boundary × weight coefficient |
| | The degree of policy and system support is much reduced | Willingness of PC to participate in URIT × weight coefficient |
| | Amount of improvement in project attractiveness | Co-operation degree of residents × weight coefficient + return on investment × weight coefficient + industrial import and operation × weight coefficient |
| | Decrease in project attraction | Change risk × weight coefficient + construction cost × weight coefficient |
| Auxiliary variables | Level of government finance | WITH the LOOKUP (Time) LOOKUP ([(0, 0)–(1, 2)], 0.695 (0), (2, 0.798542), 0.946124 (4), (6, 1.09827), 1.3048 (8), (10, 1.39)) |
| | Different types of targeted policies | WITH the LOOKUP (Time) LOOKUP ([(0, 0)–(1, 2)], 0.7 (0), (2.5, 0.875), (5, 1.05), (7.5, 1.225), (10, 1.4)) |
| | Degree of co-operation among the inhabitants | Initial value of residents' collaboration + interpretation of policy publicity × weight coefficient |
| | Industry introduction and operations | Initial value of industry import and operation + regulation of floor area ratio index × weight coefficient |
| | Risk of change | Initial value of change risk + Project construction period × weighting coefficient–Degree of resident collaboration × weighting coefficient |
| | Construction cost | Initial value of construction cost − Interest subsidy × weighting factor + Tax credit × weighting factor |

Although the realization of the system dynamics model does not solely rely on data, it is still crucial to make the study more closely aligned with the real system by incorporating some real data into the modeling process. Therefore, references to real data are used to assign values to certain equations of auxiliary variables in the model.

Under Hypothesis 3, the level of government finances will increase annually over time, following a consistent trend corresponding to the calendar year. To capture the trend of the shift during the simulation period, a tabular function was used, using statistics selected from previous years in Nanjing. The fiscal revenue levels of the Nanjing Municipal Government from 2011 to 2021 were derived from the Nanjing Municipal Statistical Yearbook (2011–2021), as shown in Table 4. Based on these data, tabular functional equations for the auxiliary variables representing the fiscal level of the government were formulated in the system dynamics model.

**Table 4.** Fiscal revenue of the Nanjing government from 2015 to 2021 (unit: CNY 100 million).

| Year of year | 2011 | 2012 | 2013 | 2014 | 2015 | 2016 |
|---|---|---|---|---|---|---|
| Fiscal Revenue | 1298.77 | 1427.25 | 1591.59 | 1771.85 | 2008.96 | 2198.54 |
| **Year** | 2017 | 2018 | 2019 | 2020 | 2021 | |
| **Fiscal Revenue** | 2439.23 | 2783.84 | 3023.3 | 3009.55 | 3264.26 | |

According to Hypothesis 4, the government gradually improved targeted policies for different types of URIT projects. The growth rate of the targeted policies for each type was determined based on the level of completeness of the targeted policies introduced by municipal governments for URIT projects. According to the Pilot Implementation Plan for URIT in Nanjing, published on 24 March 2022 by the Nanjing Urban and Rural Construction Commission, the URIT has been classified into four types: residential lot renewal, production building renovation, public space upgrading, and comprehensive area renewal. However, the current update of detailed policy documents is primarily focused on residential lots, such as the "Guiding Opinions on Carrying out URIT of Residential Lots" (Ning Planning Resources [2020] No. 339) and the "Detailed Rules for the Implementation of URIT of Residential Lots Planning and Land". Additionally, there are provisions such as the "Provisions on Further Strengthening the Wind and Landscape Control and Strictly Controlling the Height of Buildings in the Old City Planning and Management Regulations" (Ning Zheng Planning Zi [2023] No. 3) that address historic neighborhood transformation. However, for the additional three types, there is a lack of transformation approaches or specific details regarding URIT. Therefore, it can be assumed that the government will gradually refine the methodology and approach for updating the detailed regulations on URIT for these four types. This assumes that changes occur over time.

2.    Assignment of weight coefficients and initial values of variables

It is challenging to express variables in terms of real values, since most variables in the system are dimensionless. In addition, the weighting coefficients of each variable in the system are assigned based on the weights of each influence factor in the computation. Initial values were determined using the CRITIC method based on the average scores of the influencing factors. To facilitate the uniform computation of real and simulated data, a parameter normalization procedure was applied to the initial values of the system variables. The results for the weighting coefficients and initial reference volume assignments for the system variables are presented in Table 5.

**Table 5.** Weight coefficients and variable initial values.

| Name | Value | Name | Value |
|---|---|---|---|
| Weight coefficient of government overall management degree | 0.2756 | Weight coefficient of relevant policy support degree | 0.3742 |
| Project attractiveness weight coefficient | 0.3501 | Weight coefficient of government fiscal level | 0.044 |
| Co-ordinate departments to establish weight coefficients | 0.038 | Weight coefficient of open selection mechanism | 0.041 |
| Weight coefficient of government performance spirit | 0.029 | Clear weight coefficient with enterprise management boundary | 0.033 |
| Policy publicity interpretation weight coefficient | 0.029 | Approval process simplified weight coefficient | 0.033 |
| Weight coefficient of flexible conversion of land use nature | 0.034 | Property rights division clear weight coefficient | 0.036 |
| Weight coefficients of different types of targeted policies | 0.035 | The plot ratio index regulates the weight coefficient | 0.038 |
| Weight coefficient of interest subsidy | 0.038 | Tax deduction weight coefficient | 0.044 |
| Weight coefficient of financial credit support | 0.039 | Flexibly delimit the weight coefficient of land use boundary | 0.030 |
| Weight coefficient of residents' degree of collaboration | 0.038 | Investment return weight coefficient | 0.057 |
| Industry import and operation weight coefficient | 0.043 | Change the risk weight coefficient | 0.038 |
| Construction cost weight coefficient | 0.043 | | |
| Public selection of the initial value of PC mechanism | 0.7175 | Establish the initial value of the co-ordination department | 0.6850 |
| Initial value of government performance spirit | 0.7775 | Initial value of policy publicity interpretation | 0.6175 |
| The nature of land use flexibly changes the initial value | 0.6025 | Property rights are divided into clear initial values | 0.6000 |
| Approval process: simplified initial values | 0.7100 | The plot ratio index regulates the initial value | 0.7100 |
| Flexibly delimit the initial value of land boundaries | 0.7025 | Financial credit support initial value | 0.7200 |
| Tax deduction initial value | 0.7325 | Initial value of interest subsidy | 0.6825 |
| Initial value of residents' co-operation degree | 0.6700 | Initial value of industry introduction and operation | 0.5700 |
| Return on investment initial value | 0.5025 | Change the initial value of risk | 0.6200 |
| Initial value of construction cost | 0.6025 | Initial value of the project construction period | 0.6200 |

Among them, due to the elimination of the influencing factors of the "enterprise capacity" dimension, only the influencing factors of the three dimensions of "government behavior", "policy support", and "project situation" remain in the system of factors influencing in the government's perspective. Therefore, the weighting coefficients of the weighting of the three state variables for normalization, on the basis of the original weighting coefficients of the formula, are as follows:

$$y = \frac{x - x_{\min}}{x_{\max} - x_{\min}} \tag{5}$$

## 4. Results

### 4.1. Model Validation

As described in the previous sections, private capital is an important force to participate in urban and rural infrastructure transformation (URIT); thus, this paper constructs an evolutionary system dynamics model of PC's willingness to participate in URIT on the basis of constructing a system of influencing factors of PC's participation in urban regeneration investment projects, and further explores the correlation between PC's willingness to participate and each of the influencing factors. The following is an analysis of the results of the model simulation.

(1)    Visual test

The impact factor model of PC participation in URIT was tested for both structure and consistency. The "Check Model" function in the Vensim PLE x64 software was utilized for this purpose. The tests include assessing the plausibility of causality, the completeness and correctness of variable definitions, and the consistency of variable units. The results of the test were successful.

(2)    Model structure test

The structural test primarily focuses on testing the stability of the model and observing the sensitivity of parameter changes in the system to the model. To test the structural stability of the model, the state variable representing the willingness of PC to participate in URIT was selected as a test indicator. The simulation time step sizes of 0.25, 0.5, and 1 were set, and the model was run. The results are shown in Figure 5. The development in the trend of the willingness of PC to participate in URIT remains consistent across different simulation time steps, indicating that the model structure exhibits stability.

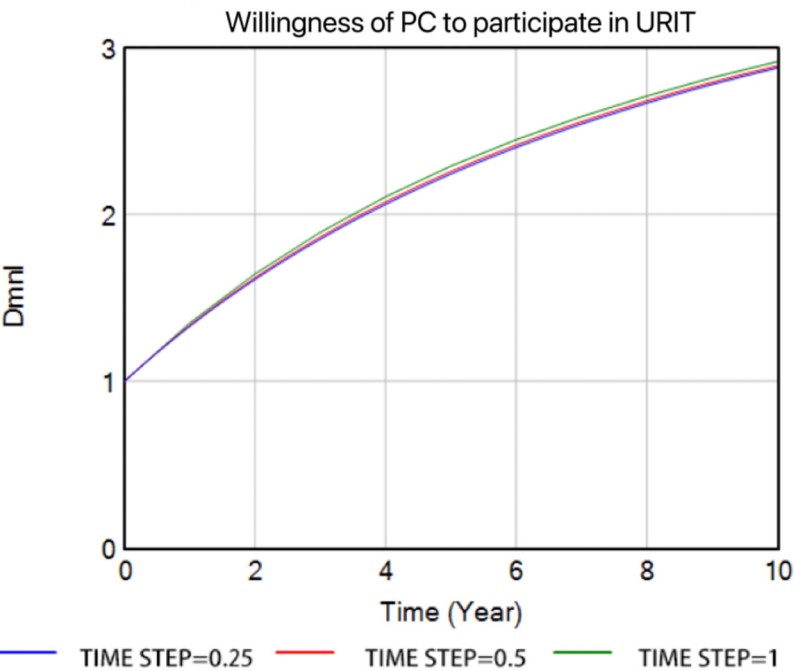

**Figure 5.** Comparison of the willingness of PC to participate in URIT with different simulation time steps.

*4.2. System Simulation and Mechanism Analysis*

4.2.1. Simulation of the Development Trend of the Baseline Scenario

The trend in the baseline scenario simulation is to simulate the system without changing the values of the various influencing factors in the system. The simulation results of the system model for the factors influencing PC participation in URIT are shown in Figures 6 and 7.

From the simulation results, it can be seen that the level of government co-ordination and management will rise over time as the level of government finances grows year by year, with a greater ability to pay for PC remuneration and a greater willingness of PC to co-operate with the city government. The attraction of PC to URIT projects will increase. This is influenced by two factors: firstly, the government's publicity and promotion efforts, which have led to a shift in the overall attitude of society towards URIT; and, secondly, the increasing degree of co-operation from the public. On the other hand, the government will continue to introduce targeted policies for different types of URIT projects. These policies will regulate renovation methods and regulations for areas such as former factories and

historical and cultural neighborhoods. Such regulations will enable PC to participate in URIT operations in accordance with established rules and regulations. The government has also implemented relevant policy measures, including incentives, to encourage PC participation by reducing the project costs associated with their involvement. As a result, there will be a corresponding increase in the willingness of PC to participate. However, this higher willingness changes over time, and will not be sustained by policy encouragement from the government. Therefore, the upward trend of the willingness of PC to participate in URIT gradually tends to level off under the combined effect of these three factors.

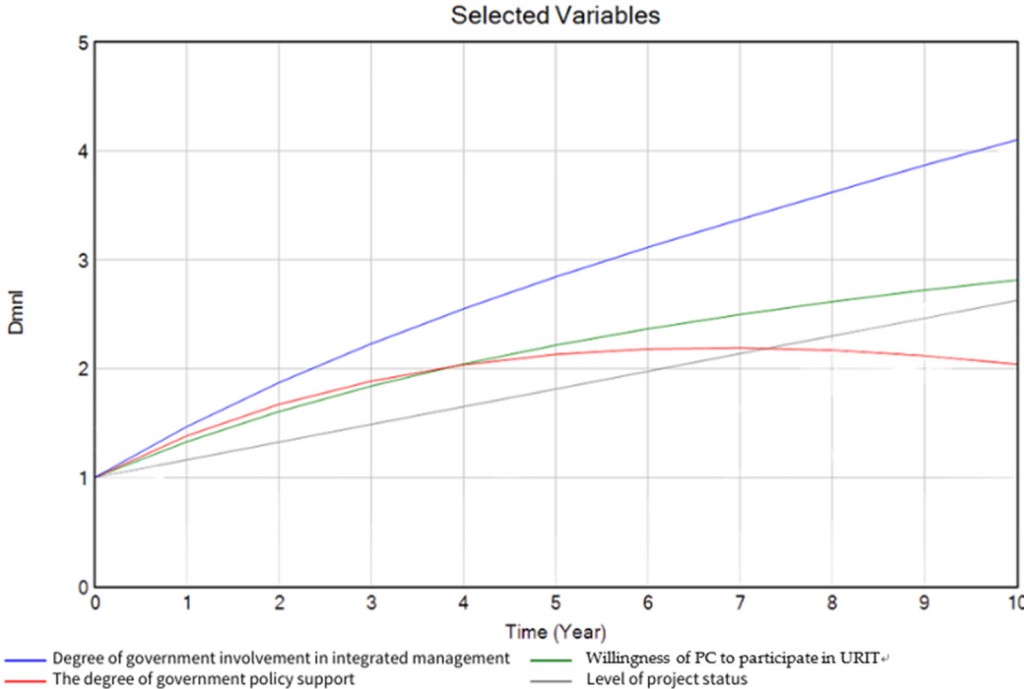

**Figure 6.** Willingness of PC to participate in URIT and the development of related influencing factors under the baseline scenario.

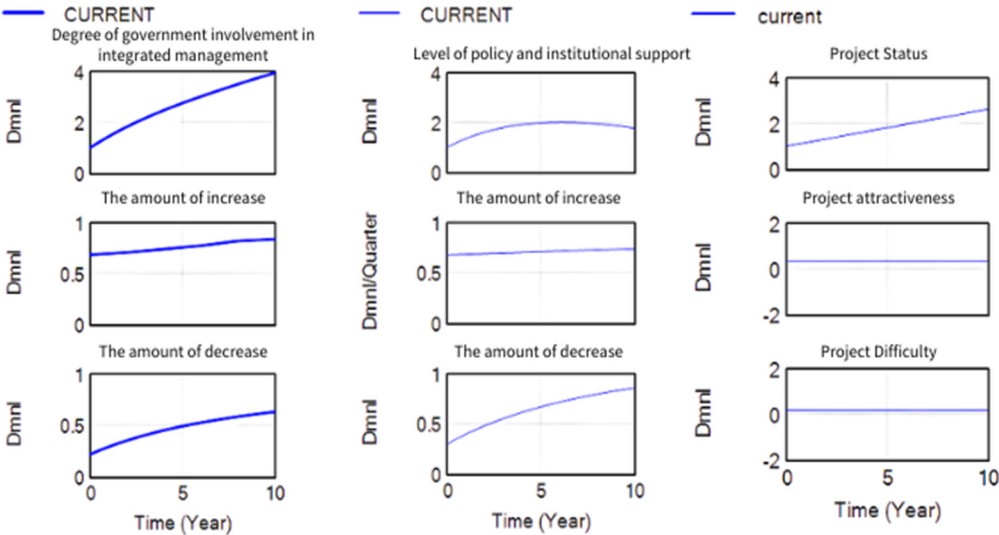

**Figure 7.** Trends of government co-ordinated management, policy and system support, and project situation.

### 4.2.2. Government Behavior Simulation and Influence Mechanism Analysis

The government acts as a project partner and coordinator in "government-led, market-participating" URIT projects. Its behavior has some impact on the willingness of PC to participate in URIT. However, due to the fact that URIT has been written overly late into China's planning, numerous city governments are still exploring the allocation of functions and mechanism processes in the area of URIT, so that sound coping mechanisms can be established. In this context, the government needs to continuously improve its function allocation and play a good leadership role.

In system dynamics, we can observe the trends of a system in a state by changing the relevant variables and simulating the effects of changing government behavior by changing the values of certain variables. Therefore, we set up five scenarios of changes in this article, namely, "the level of implementation of the public selection mechanism for PC", "the degree of coordination by government departments", "the spirit of government performance", "the degree of clarity of the management boundary between the government and enterprises", "the interpretation of policy propaganda", "the degree of clarity of the management boundary between the government and enterprises", and "policy publicity and interpretation". The degree values of the five shift scenarios were set to increase by 0.2 year on year to observe the degree of impact of their change on the willingness of PC to participate in URIT. The results are shown in Figure 8, where the current scenario is the original state without adjusting any variables.

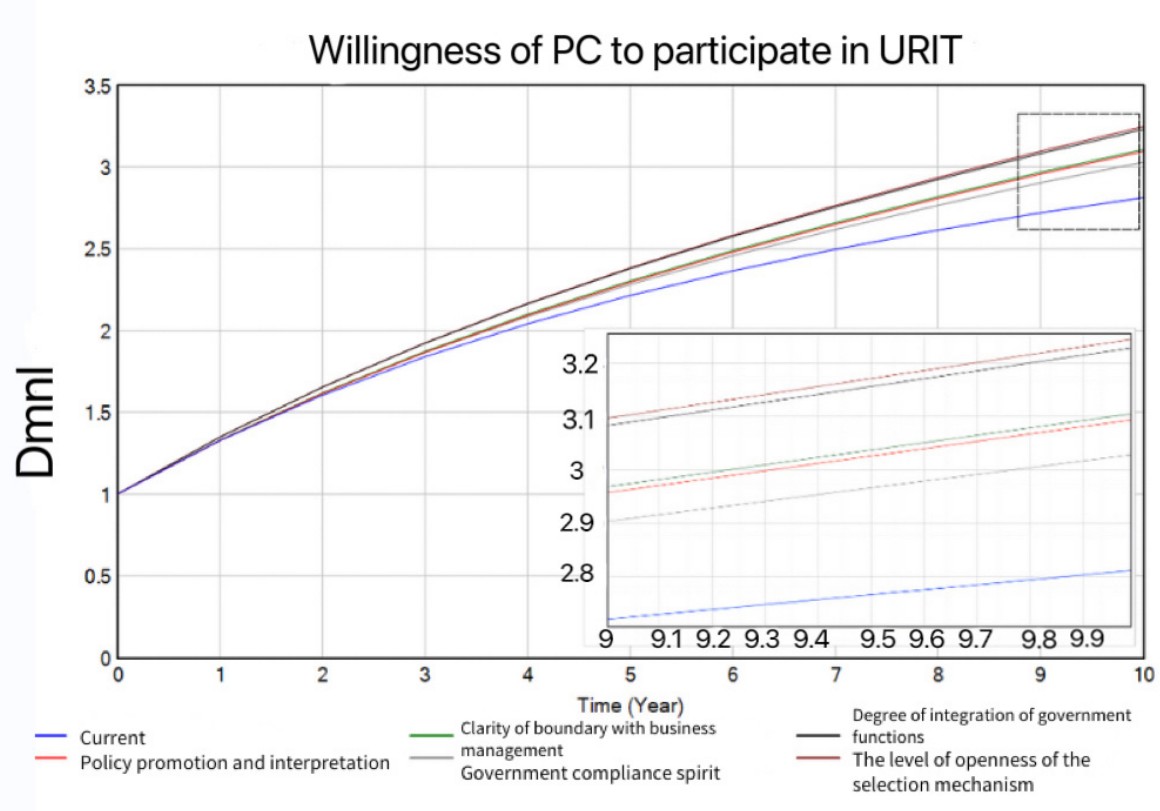

**Figure 8.** Comparison of the willingness of PC to participate in URIT under different levels of government management.

The willingness of PC to participate in URIT is enhanced compared to the original state when the level of each variable increases. Each variable has a facilitating effect on the willingness of PC to participate in URIT, while the level of the impact varies. In descending order, they are "open selection mechanism for PC", "co-ordination of government

departmental functions", "clear management boundary with enterprises", "policy dissemination", "interpretation", "policy support", "interpretation", and "spirit of government performance".

Among them, the shift in the two factors of "public selection of PC mechanism" and "co-ordination of government departments' functions" has the highest degree of influence on the willingness of PC to participate in URIT, and the influence effects of the two factors are comparable. Under the influence of these two factors, the willingness of PC to participate in URIT has increased by about 40 percent compared to the original scenario. The relative values of the willingness of PC to participate in the baseline scenario at the end of the simulation time were raised by 2.81 to 3.24 and 3.22, respectively. Two major problems are reflected in the participation of PC in URIT projects. One is that there is no clear mechanism for selecting PC to participate in URIT for co-operation, which leads the government to choose familiar state-owned enterprises based on past experience. Private enterprises that have the capital and technology find it difficult to enter into URIT. Another problem is the fragmentation of government departmental functions. The long-standing governance of sectoral classification and the lack of a dedicated department for integrated management have led to the emergence of multiple leaders and complex approval issues in the process of PC participation in URIT.

The three influencing factors of "clear management boundaries with enterprises", "policy propaganda and interpretation", and "the spirit of government compliance" have a relatively low impact on the willingness of PC to participate. However, their role in raising the level of PC participation should not be ignored. Among them, the role of policy publicity and interpretation in improving the willingness of PC to participate in the transformation is shown in two respects: on the one hand, it directly affects the PC by making the current introduction of the policy system and regulations, the operation process, the specific preferential methods, and the scope of application and other content more precisely understood. This will increase the incentive for PC to participate in URIT. On the other hand, it acts on the overall social environment to improve the understanding and perception of communities and residents towards URIT, and to create an open and inclusive environment for URIT across society. Improving the level of co-operation among residents in terms of URIT reduces the risk that the process of PC participation in transformation may encounter social risks to some extent.

### 4.2.3. Policy System Simulation and Influence Mechanism Analysis

The policy system is the main means by which the government incentivizes the participation of PC in URIT. At present, the research on URIT has become mature, but the participation of PC in URIT is not extensive. Most of them discuss URIT from the perspective of the government. Chen [50] bridges the digital divide between urban and rural areas from the perspective of the government to promote the transformation of the dual economic structure; Liu et al. [51] extended the concept of urban and rural infrastructure to make it closely integrated with the progress of economic society and technological innovation. Based on the two critical application scenarios of new urbanization and rural revitalization, Gao et al. [52] organized the opportunities and challenges faced by China's current development according to three aspects: urban and rural digital divide, new consumption, and new demand for safety and health. Therefore, this paper constructs a system of factors affecting PC's participation in URIT, adopting a combination of the analytic hierarchy process (AHP) and criterion importance through criterion correlation (AHP-CRITIC) methods to quantify the comprehensive assignment of influencing factors, which has certain advantages compared with existing relevant work. In recent years, central and local governments have issued a large number of policy documents related to URIT, some of which have dealt with the participation of PC in URIT, but the results have not met expectations. Since the degree of support for the policy regime is limited, it is not possible to keep increasing variables such as the degree of volume rate relaxation or the level of financial support, and at the same time, it is not possible to gradually raise the level year by year, such as the influencing

factor of government behavior. Thus, in this section, the initial value of each influence variable is doubled at the level of the policy system and the initial values of the other variables are unchanged. The system runs and changes the situation of the willingness of PC to participate in URIT as each variable is increased, as shown in Figure 9.

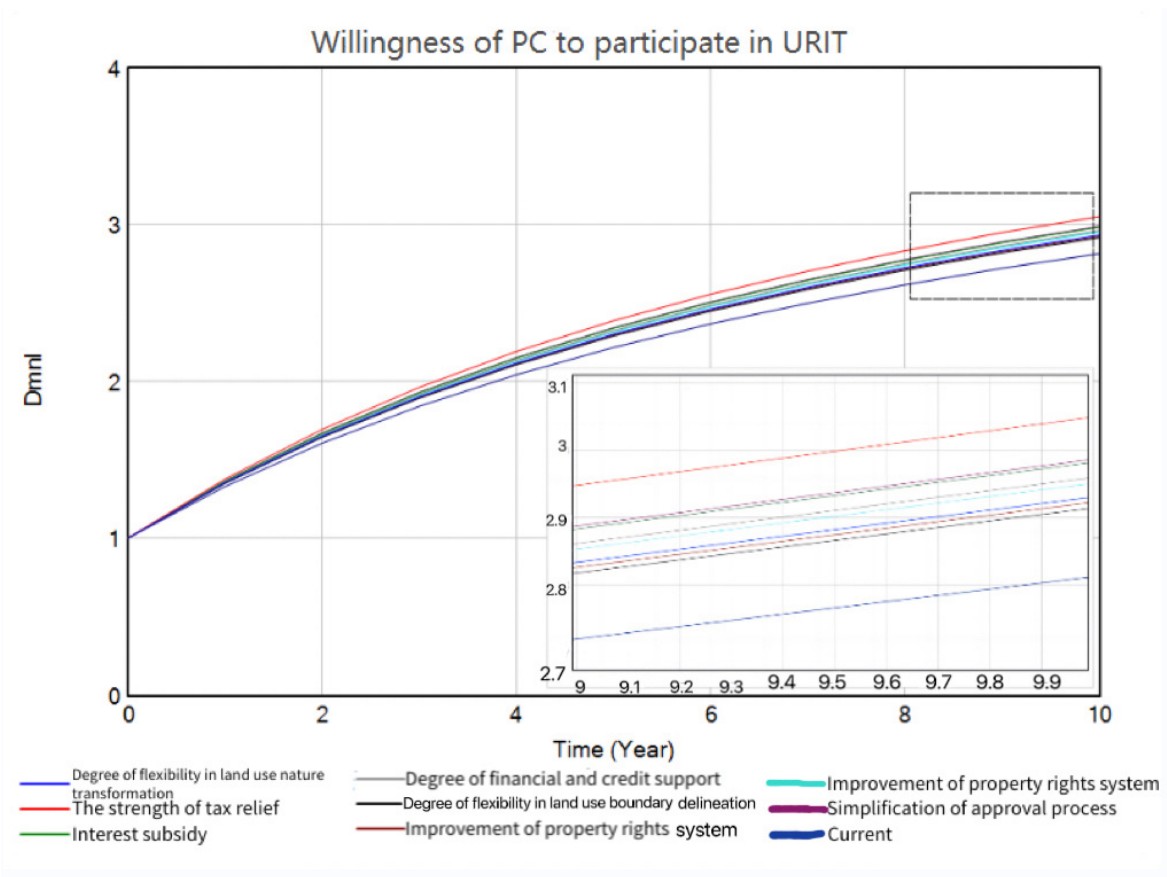

**Figure 9.** Comparison of the willingness of PC to participate in URIT under different policy system support scenarios.

The simulation results show that doubling the tax exemption level has the highest impact on the willingness of PC to participate in URIT. It increases the relative value of the baseline scenario's PC participation willingness from 2.81 to 3.05 by the end of the simulation period and reduces the time for PC's willingness to participate by one year compared to the initial time. "Regulation of floor area ratio", "interest subsidy", "financial credit support", and "simplification of approval process" are second, and "flexible transformation of land nature", "improvement of property rights system", and "flexible delineation of land boundary" have a relatively low impact.

It can be seen that the issue of finance remains a major concern for PC when contemplating participation in micro-profitable URIT projects. PC is more willing to participate in traditional new construction projects and demolition and reconstruction types of URIT projects due to the elevated cost and low return of the project itself, with fewer being willing to participate in micro-profit URIT projects. However, fiscal policy support such as tax reductions and interest subsidies can directly reduce project construction costs to a certain extent and increase the rate of the return on investment of the project, which is in line with the profit-driven nature of PC in the private sector. Another approach to increasing the return on investment is through the relaxation of plot ratio targets. By allowing projects to have a higher floor area, the plot ratio of the land can be adjusted to a certain extent. For example, this can involve raising buildings in ancient neighborhoods or reorganizing land in urban villages. The additional floor area obtained through this relaxation can be

used for transfer, sale, or to introduce tertiary industries for subsequent operation. This helps maintain the financial balance of the project. However, it should be noted that the relaxation of the plot ratio objective inevitably raises design issues related to the tenure of the current building area. Therefore, there is a need to improve the property rights system.

## 5. Discussion

### 5.1. Countermeasures and Suggestions for Governmental Encouragement of PC Participation in URIT

Exploring the participation of PC has become an important aspect of URIT. Shen et al. [53] changed the transformation logic from land capital-driven spatial production to private capital-driven community construction. The need to explore the evolution of PC from various actors in the URIT process has been acknowledged, including factors such as government–enterprise co-operation and community participation [54]. This article innovatively identifies the influencing factors and constructs incentivizing strategies to help improve PC's participation in urban renewal, providing a continuous impetus for URIT and the transformation of urban and rural governance. Based on the analysis process described above, this paper presents targeted strategies and recommendations for governments to motivate PC to participate in URIT. These strategies focus on top-level design, synergy, and resource allocation, as follows:

1. Improve the completeness of supporting policies and strengthen the cohesion of territorial policies.

The higher impact of the policy regime on the willingness of PC to participate in URIT has been identified through influencing factors and simulations of system dynamics. The "1 + N" policy system of advanced cities should be properly learned from in order to improve the coverage and compatibility of policies. This includes establishing a "policy toolbox" for URIT. Additionally, there should be a focus on continuity and convergence between old and current policies, as well as the integration of policies and their regional implementation. The government should play a leading role in all aspects of URIT, providing a favorable policy and institutional environment for PC to participate in market-based URIT, thus stimulating the motivation of PC to participate.

2. Give play to the leading role of government functions and improve the degree of co-operation between government and enterprises.

The government demonstrates strong leadership and macro-control capabilities in urban construction. In the government-led and market participation model of URIT, maintaining the sustainability of URIT projects, especially those with a micro-profitable nature, requires the government to take the lead. This entails creating a more inclusive, open, and favorable environment for URIT in society as a whole, thereby enhancing the motivation for PC participation.

3. Strengthen the updated project profit point and deepen the participation of PC.

Based on the analyses of the impact mechanisms in Section 5.2, it is evident that the financial aspect remains a primary concern for PC participation in micro-profitable URIT projects. The return on investment of the project carries significant weight in the overall impact factors. Therefore, the key to promoting micro-profitable URIT projects lies in strengthening their profitability.

### 5.2. Limitations and Future Research

Due to the limited knowledge in the application domain and the continuous development of new technologies, the research presented in this paper still needs to be further considered for optimization and improvement in terms of analysis scenarios and technical links, as follows:

1. It is possible that the limitations of the information gathered in the survey on the impact factors of PC participation in URIT may have affected the completeness of

the selected indicators. As a next step, it would be beneficial to expand the coverage of the questionnaire to collect more samples and to consider using methods such as big data text processing to identify additional influencing factors. This will help to optimize the scientific and comprehensive identification of the influencing factors.

2.  The timely follow-up, analysis, and improved evaluation of the impact of future incentive policy implementations are necessary to further enhance the willingness of PC to participate in URIT. It is also important to validate the analysis of the system dynamics model simulation results through long-term real-world cases and professionally analyzed evaluation methods. Therefore, further exploration is needed to investigate the implementation of sustainable adaptive optimization mechanisms for incentive policies aimed at encouraging PC participation in URIT.

## 6. Conclusions

In this paper, we developed a systematic dynamical model of PC participation in URIT. We conducted two simulations to explore the effects of government behavior and policy regimes. Based on our analysis, the following conclusions can be drawn:

1.  Based on the analysis of policy texts and the research literature, this article identified 27 major factors influencing PC participation in URIT. These factors were categorized into four dimensions: government behavior, policy regime, project situation, and enterprise capacity. This identification and empowerment of the influencing factors aimed to address the challenges within the framework of government–enterprise collaboration. Based on the portfolio weight level analysis, it was found that, among the 27 influencing factors, "project investment return", "tax incentives", "government fiscal level", "project construction costs", "post-project introduction and operation", "enterprise financing capabilities", "enterprise financial situation", "open selection mechanism for PC", "financial institution credit support", and "establishment of a special aggregate department" have the greatest impact on PC participation in URIT. In contrast, factors such as "enterprise technological level", "government performance spirit", "enterprise social responsibility", and "policy promotion" have a relatively insignificant impact on PC participation in URIT.

2.  In order to analyze the logic mechanism of PC participation in URIT, this paper constructed a system dynamics model. This model incorporated the identified influencing factors and their corresponding weighting processes. By doing so, the changing trends of PC's willingness to participate under the influence of each factor were identified. By analyzing the simulation results, it was observed that increasing the degree of implementation of "public selection of PC" and "establishment of co-ordination departments" among the influencing factors related to the government's behavior significantly enhanced the willingness of PC to participate in the final stage of the simulation. The willingness increased from 2.81 to 3.24 and 3.22, respectively. Furthermore, after doubling the "tax relief" within the policy system, the willingness of PC to participate increased from 2.81 to 3.05 in the final simulation.

3.  In order to construct a sustainable incentive strategy for PC participation in URIT, this article combined the integrated application of stakeholder theory and incentive theory. Based on the analysis of the model simulation results, the article proposed a strategy for the government to incentivize PC participation in URIT. The strategy focuses on three aspects: top-level design, synergy and co-operation, and resource allocation. The proposed strategies include the following: to strengthen the profitability of URIT projects, it is essential to innovate the mode of participation of PC. This can be achieved by ensuring the completeness of support policies and enhancing the articulation of territorial policies. Additionally, the government should play a leading role in facilitating government–enterprise co-operation and enhancing their compatibility.

These findings can provide operable strategies and measures for the government to guide URIT work in a market-oriented way and formulate relevant incentive policies, which can help the government to use the power of social capital for urban renewal construction

more effectively, and also encourage social capital's participation in the urban renewal work in a more unimpeded way, so as to realize a win–win situation between the government and social capital, to push forward the process of China's urban renewal, and, ultimately, to promote the high-quality development of China's cities. This paper does not only provide guidance for technical methods and scientific ideas for real work, but also has the following originality:

1. The research establishes the influencing factors involved in PC participation in URIT, systematically organizing them and arriving at a more reasonable weighting of the influencing factors through the AHP-CRITIC combination method. This approach provides solid theoretical references for exploring the effect of each influencing factor on the willingness of PC participation.
2. The research also establishes a model of the influencing factors of PC participation in URIT and uses system dynamics to illustrate the abstracted current situation of PC's willingness to participate in URIT and the development trend of change in a concrete way. Moreover, the work analyzes the effect of each influencing factor on PC's willingness to participate in URIT.

The focus of this paper was to address the problem of insufficient willingness of PC to participate in URIT programs. The series of measures and policy recommendations proposed by existing studies to address this issue are too general, favoring directional guidelines, and more detailed operational guidelines are still needed for practical applications. Instead, this paper aimed to enhance the framework of systematic and targeted incentive strategies for the government to encourage PC's engagement in URIT. The findings of this research provide strong support for the government in optimizing the pathway of URIT through effective incentives. Furthermore, this work lays a solid theoretical and practical foundation for future research on systematic incentive strategies for URIT.

**Author Contributions:** Conceptualization, H.C. and G.F.; methodology, H.C. and Y.Z.; software, Y.Z. and H.Y.; validation, Y.Z. and H.C.; formal analysis, X.D. and Y.Z.; investigation, Y.Z., H.C. and G.F.; writing—original draft preparation, H.C., Y.Z., X.D. and H.Y.; writing—review and editing, H.C., X.D., H.Y. and G.F.; supervision, G.F. All authors have read and agreed to the published version of the manuscript.

**Funding:** This research was funded by the Anhui Philosophy and Social Science Planning Project, grant number "AHSKY2020D08".

**Data Availability Statement:** The data analysis results presented in the paper are available from the corresponding author on request.

**Conflicts of Interest:** The authors declare no potential conflicts of interest with respect to the research, authorship, and/or publication of this article.

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
