# Peer review of "Identification and Simulation of the Influencing Factors of Private Capital Participation in Urban and Rural Infrastructure Transformation Based on System Dynamics"

_buildings, doi:10.3390/buildings13092327_

Round 1
Reviewer 1 Report
The authors evaluate the "Identification and Simulation of the Influencing Factors of Private Capital Participation in Urban and Rural Infrastructure Transformation Based on System Dynamics". It is an interesting and significant contribution to the scientific community; however, some errors in the manuscript should be clearly discussed, and the manuscript needs some revisions, as
given below:
Comments -
1. English should be enhanced.
2. What is the originality of this study?
3. Figures 2, 3, and 4 - readability is very low; enhance them.
4. Separate the "4. Analysis and Results" into two sections. Otherwise, understanding this study is challenging. Include the Analysis part in the methodology section.
5. Enhance the discussion with other related studies.
6. The paper's research design, including the study method and justifications, should be clearly presented to evaluate the scientific validity of the research.
7. "3.2. System Dynamic Model" - missing references. Check the whole manuscript and revise them where necessary.
Moderate editing of English language required
Author Response
Response to Reviewer 1 Comments
We would like to thank you for your careful reading, helpful comments, and constructive suggestions, which has significantly improved the presentation of our manuscript. We have carefully considered all comments from the reviewers and revised our manuscript accordingly. The manuscript has also been reviewed, the syntactic and grammatical errors we found have been corrected, and the content has been supplemented. In the following section, our responses are given in red font and changes/additions to the manuscript are given in blue text. We believe that our responses have well addressed all concerns from the reviewers. We hope our revised manuscript can be accepted for publication.
Point 1: English should be enhanced.
Response 1: We sincerely appreciate the valuable suggestion. However, we do invite a friend of us who is a native English speaker from the USA to help polish our article. And we hope the revised manuscript could be acceptable for you.
Point 2: What is the originality of this study?
Response 2: Thank you for your valuable suggestions. Based on your comments, we have revised the "Conclusion" section of the manuscript by reorganizing the article. On pages 26-27 (line 864-872) of the Conclusion, we have added the following information about the originality of this study:
- Establishing the influencing factor system of PC participation in URIT, systematically sorting out the influencing factors, and arriving at a more reasonable weighting of the influencing factors through the AHP-CRITIC combination method, which provides solid theoretical references for exploring the effect of each influencing factor on the willingness of PC participation.
- To establish a model of the influencing factors of PC participation in URIT, to use system dynamics to show the abstracted current situation of PC's willingness to participate in URIT and the development trend of change in a concrete way, and to analyze the effect of each influencing factor on PC's willingness to participate in URIT.
Point 3: Figures 2, 3, and 4 - readability is very low; enhance them.
Response 3: We were really sorry for our careless mistakes. Thank you for your reminder. As suggested by the reviewer, we have redrawn Figures 2, 3 and 4 to enhance readability. Due to some adjustments in the article chapters, the original Figure 2, 3 has been changed to Figure 3, 4, and the original Figure 4 has been deleted.
Point 4: Separate the "4. Analysis and Results" into two sections. Otherwise, understanding this study is challenging. Include the Analysis part in the methodology section.
Response 4: We gratefully appreciate for your valuable comment. According with your advice, we changed the fourth section from “Analysis and Results” to “Results”, retained the part after 4.2.4. Model Validation, and added a beginning paragraph in lines 605-611 to highlight the research significance and make the context more coherent. And the previous content was merged into methodology. According to the logical relationship between the merged content and the original content in Chapter 3, the order is adjusted. Our specific additions to the new section 4.1 Model Validation are as follows:
As it has been known in the previous chapters that private capital is an important force to participate in urban and rural infrastructure transformation (URIT), this paper constructs an evolutionary system dynamics model of PC's willingness to participate in URIT on the basis of constructing a system of influencing factors of PC's participation in urban regeneration investment projects, and further explores the correlation between the PC's willingness to participate and each of the influencing factors, and the following is an analysis of the results of the model simulation:
Point 5: Enhance the discussion with other related studies.
Response 5: Thank you for your good suggestion. As suggested by the reviewer, we added the part citing the importance of PC in lines 762-764 of 5.1. Specific modifications are as follows:
Exploring the participation of PC has become an important idea of URIT. Shen et al. [51] change the transformation logic from land capital-driven spatial production to private capital-driven community construction.
Point 6: The paper's research design, including the study method and justifications, should be clearly presented to evaluate the scientific validity of the research.
Response 6: We think this is an excellent suggestion. We have added this section 3.1. Research design at the beginning of Chapter 3 on pages 6-7 which introduced the selection of research methods and research ideas in detail in accordance with your suggestion. In view of this problem, the research design of this paper is as follows:
In the transformation of urban and rural stock, the transformation of infrastructure cannot be sustained only by the limited financial power of the government. In areas that are in urgent need of transformation but where the rate of return on trans-formation is low and implementation is relatively difficult, it is often necessary to guide the flow of market capital through incentives, and how to make flexible use of incentive strategies to fully utilize market resources in the process, to adjust the relationship between land property rights, to balance the interests of all parties, and to improve the efficiency of urban resource allocation, has become a core issue that urgently needs to be explored and resolved.
Therefore, from the perspective of efficient government regulation, in view of the current status quo of low motivation of PC participation in URIT, we identify the influencing factors of PC participation in URIT and determine the influencing weights, and establish the system dynamics evolution architecture of PC participation in URIT. Through software simulation analysis, the article observes the change development trend of PC's willingness to participate in URIT under the role of various influencing factors, and accordingly puts forward targeted countermeasures and suggestions for government to incentivize PC's participation in URIT. In this way, the government can solve the practical problems such as insufficient consideration of incentive strategies and unclear participation mechanisms that exist when encouraging PC to participation in URIT and developing market-oriented transformation, and thus to promote the process of government-led and market-operated URIT.
In this paper, literature analysis is used to sort out and summarize the relevant literature about URIT, and on this basis, the research results on the influencing factors of PC participation in URIT in various literatures are statistically summarized. Then, through the combination of AHP and CRITIC analysis, subjective and objective judgment are made on the weights of the influencing factors of PCs participation in URIT, and the empirical knowledge and abstract judgment of experts are visualized for data processing. Finally, on the basis of system dynamics, we construct a model of factors influencing the change of PC's willingness to participate in URIT, so as to more intuitively describe the process of the dynamic change of its willingness, and provide a model basis for the construction of the government's strategy for encouraging PC's participation in URIT.
Point 7: "3.2. System Dynamic Model" - missing references. Check the whole manuscript and revise them where necessary.
Response 7: We were really sorry for our careless mistakes. Thank you for your excellent suggestion. According to Reviewer's suggestion, we have added relevant literature [49] and [50] in new 3.4.1 (original 3.2). The ambiguity caused by improper expression in lines 4 to 5 of paragraph 2 of 3.2 shall be amended as follows:
Correct the original sentence“A previous paper has conducted a qualitative analysis of the factors influencing PC participation in URIT. ” to “In the previous part of this paper, the factors affecting PC participation in URIT have been qualitatively analyzed. ”
We tried our best to improve the manuscript and made some changes marked in blue in revised paper which will not influence the content and framework of the paper. We appreciate for Editor/ Reviews’ warm work earnestly, and hope the correction will meet with approval. Once again, thank you very much for you comments and suggestions.
Reviewer 2 Report
1) Numeric result is missing in Abstract. It is good. Mention the best-performing results (Quantitative values).
2) The novelty of this paper is not presented well. Please add. Contribution is missing.
3) Introduction should provide more background on the work with scope of the work
4) Contribution should be clearly identified and presented under the Introduction.
5) The paper, does not link well with recent literature on top-tier journals and research gap should be clearly identified.
6) A high-level block diagram of the entire technical work can be added at the beginning of Section 3.
7) Figure 3 should be more clearer. The equation number is missing in the entire mansucript.
8) The visual test system is limited. How this message come as “Message form vensim”. Please add details about this.
9) A performance/advantages comparison with existing related works should be added at the end of the result section to validate the proposed method’s capability.
10) Results section should be updated by adding the strength, and impact/significance of this work in real-life scenarios.
Minor
Author Response
Response to Reviewer 2 Comments
On behalf of all the contributing authors, I would like to express our sincere appreciations of your letter and reviews’ constructive comments concerning our article entitled “Identification and Simulation of the Influencing Factors of Private Capital Participation in Urban and Rural Infrastructure Transformation Based on System Dynamics”. These comments are all valuable and helpful for improving our article. We have carefully considered your comments and revised our manuscript accordingly. The manuscript has also been reviewed, the syntactic and grammatical errors we found have been corrected, and the content has been supplemented. In the following section, our response is given in red font and changes/additions to the manuscript are given in the blue text. We believe that our responses have well addressed all concerns from the reviewers. We hope our revised manuscript can be accepted for publication.
Point 1: Numeric result is missing in Abstract. It is good. Mention the best-performing results (Quantitative values).
Response 1: Thank you for your excellent suggestion. In the second half of the “Abstract”, we addded the simulation results of a system dynamics model that embodies the quantization process, which combined the identified influencing factors and their corresponding weighting processes. The specific modifications are as follows:
By analyzing the simulation results, it was observed that increasing the degree of implementation of "public selection of PC" and "establishment of coordination departments" among the influencing factors related to the government's behavior significantly enhanced the willingness of PC to participate in the final stage of the simulation. The willingness increased from 2.81 to 3.24 and 3.22, respectively. Furthermore, after doubling the "tax relief" within the policy system, the willing-ness of PC to participate increased from 2.81 to 3.05 in the final simulation time.
Point 2: The novelty of this paper is not presented well. Please add. Contribution is missing.
Response 2: We sincerely appreciate the valuable comments. We find your opinion very useful. In response to this problem, we added originality to the “Conclusion” section of the paper lines 864-872, confirming the novelty of our study. The specific content we added is as follows:
- Establishing the influencing factor system of PC participation in URIT, systematically sorting out the influencing factors, and arriving at a more reasonable weighting of the influencing factors through the AHP-CRITIC combination method, which provides solid theoretical references for exploring the effect of each influencing factor on the willingness of PC participation.
- To establish a model of the influencing factors of PC participation in URIT, to use system dynamics to show the abstracted current situation of PC's willingness to participate in URIT and the development trend of change in a concrete way, and to analyze the effect of each influencing factor on PC's willingness to participate in URIT.
Point 3: Introduction should provide more background on the work with scope of the work.
Response 3: Thank you for your excellent suggestion. According with your advice, we amended the “Introduction” part in manuscript. The introduction of the work background in our original manuscript was not detailed enough. In response to this problem, we added more policy and environmental background on "Urban and rural infrastructure transformation" to make it more convincing, specifically as follows:
Since the reform and opening up, the development of Chinese cities has been mainly based on the outward expansion of urban boundaries, and the hard constraint on spa-tial resources has become the bottleneck for the development of most cities. The tradi-tional crude extensive incremental expansion mode can no longer meet the needs of today's social development, and the main battlefield of urban development is gradual-ly turning to URIT. …… The endowment of a city's hard infrastructure may not play a decisive role in enhanc-ing competitiveness, whereas social infrastructure promotes well-being and improves the quality of the population, thereby enhancing the performance of urbanization [5].
In 2023, the Ministry of Housing and Urban-Rural Construction of China issued a document pointing out that urban design should be regarded as an important means of URIT, improve the urban design management system, specify the design requirements for buildings, districts, communities, blocks, cities and villages at different scales, pro-pose the design conditions for the construction and transformation of urban and rural infrastructure plots, and organize the preparation of key project design plans in order to regulate and guide the implementation of URIT projects.
Point 4: Contribution should be clearly identified and presented under the Introduction.
Response 4: We gratefully appreciate for your valuable comment. We carefully examined the manuscript and briefly supplemented the theoretical as well as practical contributions to the study in lines 117-123 of the “Introduction”.
This paper addresses the study of PC participation in URIT, broadens the research content of the new participation architecture model of PC in sustainable urban and rural development, provides effective theoretical support for further improving the systematic countermeasures and suggestions of URIT, and provides a dynamic simula-tion evolutionary path analysis paradigm for the quantitative assessment of the will-ingness of PC participation in URIT. In reality, it helps to promote the government to better utilize the power of PC for URIT and achieve a win-win situation between the government and PC.
Point 5: The paper, does not link well with recent literature on top-tier journals and research gap should be clearly identified.
Response 5: Thank you for your excellent suggestion. As suggested by the reviewers, we added relevant links to recent top journals in lines 764-769 of the first paragraph in 5.1. By mentioning the content of this paper and the previous literature, the research gap is shown. The specific amendments are as follows:
The necessity of exploring the evolution of PC of various actors in the URIT process has been paid attention to, including factors such as government-enterprise cooperation and community participation [52]. The article innovatively addresses the identification of influencing factors and the construction of incentive strategies for PC's participation in urban renewal, providing continuous impetus for URIT and the transformation of urban and rural governance.
Point 6: A high-level block diagram of the entire technical work can be added at the beginning of Section 3.
Response 6: We think this is an excellent suggestion. In response to this problem, we have improved some of the contents of the article research, added 3.1 Research design section, and added a research design roadmap in 3.1 to make the research design of this article more clear.
Point 7: Figure 3 should be clearer. The equation number is missing in the entire manuscript.
Response 7: We were really sorry for our careless mistakes. Thank you for your reminder. As suggested by the reviewer, we redrew the original Figure 3 (now changed to Figure 4) to improve the readability of the picture and numbered the equations present in the article.
Point 8: The visual test system is limited. How this message come as “Message form vensim”. Please add details about this.
Response 8: We sincerely appreciate the valuable comments. We carefully reviewed the manuscript and removed the original Figure 4 to represent the visual test results in words.
Point 9: A performance/advantages comparison with existing related works should be added at the end of the result section to validate the proposed method’s capability.
Response 9: Thank you for your excellent suggestion. According to your comments, we have carefully examined the differences between the performance of this paper and existing related work, and have analyzed and summarized this. To address this issue, we have added lines 873-878 on page 27 of the “Conclusion” as follows:
The focus of this paper is to address the problem of insufficient willingness of PCs to participate in URIT programs. The series of measures and policy recommendations proposed by existing studies to address this issue are too general, favoring directional guidelines, and more detailed operational guidelines are still needed to apply them to practice. Instead, this paper aimed to enhance the framework of systematic and targeted incentive strategies for the government to encourage PC's engagement in URIT.
Point 10: Results section should be updated by adding the strength, and impact/significance of this work in real-life scenarios.
Response 10: We sincerely appreciate the excellent comments. In order to be more convincing, we have updated the conclusion, adding the practical significance of this work in lines 855-862. The details are as follows:
Based on the results of the above research, it can provide operable strategies and measures for the government to guide URIT work in a market-oriented way and formulate relevant incentive policies, which can help promote the government to better use the power of PC for URIT, and also help the PC to participate in the URIT work in a more unimpeded way, so as to realize a win-win situation between the government and the PC, to push forward the process of China's URIT, and ultimately to promote the high quality development of China's cities.
We tried our best to improve the manuscript and made some changes marked in blue in revised paper which will not influence the content and framework of the paper. We appreciate for Editor/ Reviews’ warm work earnestly, and hope the correction will meet with approval. Once again, thank you very much for you comments and suggestions.
Round 2
Reviewer 1 Report
The paper is suitable for publication.
Author Response
Response to Reviewer 1 Comments
On behalf of all the contributing authors, I would like to express our sincere appreciations of your letter and reviews’ constructive comments concerning our article entitled “Identification and Simulation of the Influencing Factors of Private Capital Participation in Urban and Rural Infrastructure Transformation Based on System Dynamics”. These comments are all valuable and helpful for improving our article. We have carefully considered your comments and revised our manuscript accordingly. The manuscript has also been reviewed, the syntactic and grammatical errors we found have been corrected, and the content has been supplemented. In the following section, our response is given in red font and changes/additions to the manuscript are given in the green text. We believe that our responses have well addressed all concerns from the reviewers. We hope our revised manuscript can be accepted for publication.
Point 1: Does the introduction provide sufficient background and include all relevant references? (Can be improved)
Response 1: Thank you for your valuable advice. According to your suggestion, we added relevant background and information in the second paragraph of the introduction and the first sentence of the third paragraph.
That is, " URIT, brings improvement to the existing urban areas, which is a sound approach to cope with urban decay and achieve multiple socioeconomic goals [6]." is added to the first sentence of the second paragraph of the introduction. The first sentence of the third paragraph of the introduction is added with " In China, PC participation in URIT is challenging due to its unique market system and social culture [7]." Moreover, the serial number of the relevant references cited in the following paper has also been modified.
Point 2: Are all the cited references relevant to the research? (Can be improved)
Response 2: We sincerely appreciate the valuable comments. According to your comments, we examined the manuscript carefully, and made changes to some of the references. We deleted original references [1], [7], [14], and [37] from the manuscript, and replaced original reference [29] from "Haase D; Güneralp B; Dahiya B; Bai X; Elmqvist T. Global urbanization. Urban Planet Knowl Sustain Cities 2018, 19, 326-339." to "[26] Gilbert, M. R., Eakin, H., & McPhearson, T. The role of infrastructure in societal transformations. Current Opinion in Environmental Sustainability, 2022,57(C).". At the same time, we have corrected the change in the reference serial number involved.
We tried our best to improve the manuscript and made some changes marked in blue in revised paper which will not influence the content and framework of the paper. We appreciate for Editor/ Reviews’ warm work earnestly, and hope the correction will meet with approval. Once again, thank you very much for you comments and suggestions.

Reviewer 2 Report
I would like to thank the authors for addressing my comments. However, I am wondering why the last block is data set analysis in Figure 2. Please double-check.
In addition, performance/advantages comparison with existing related works should be improved further with more reference support at the end of the result section.
Author Response
Response to Reviewer 2 Comments
We would like to thank you for your careful reading, helpful comments, and constructive suggestions, which has significantly improved the presentation of our manuscript. We have carefully considered all comments from the reviewers and revised our manuscript accordingly. The manuscript has also been reviewed, the syntactic and grammatical errors we found have been corrected, and the content has been supplemented. In the following section, our responses are given in red font and changes/additions to the manuscript are given in green text. We believe that our responses have well addressed all concerns from the reviewers. We hope our revised manuscript can be accepted for publication.
Point 1: I would like to thank the authors for addressing my comments. However, I am wondering why the last block is data set analysis in Figure 2. Please double-check.
Response 1: Thank you for your reminder. We sincerely appreciate the valuable suggestion. We have carefully reviewed the manuscript and found that the original "Data set analysis" is not appropriate, and according to your suggestion, we have redrawn Figure 2., modifying the last block to "Trend prediction and mechanism analysis". Thank you again for your advice, which has enabled us to revise the manuscript.
Point 2: In addition, performance/advantages comparison with existing related works should be improved further with more reference support at the end of the result section.
Response 2: Thank you for your correction. In section 4.2.3 of "Policy system simulation and influence mechanism analysis", we have consulted some literatures, pointed out the advantages of this paper compared with the performance/advantages of existing relevant works, and provided literature references. Here are the changes:
At present, the research on URIT has become mature, but the participation of PC in URIT is not extensive. Most of them discuss URIT from the perspective of the government. Chen [51] bridges the digital divide between urban and rural areas from the perspective of the government to promote the transformation of the dual economic structure; Liu et al. [52] extended the concept of urban and rural infrastructure to make it closely integrated with the progress of economic society and technological innovation; Based on the two critical application scenarios of new urbanization and rural revitalization, Gao et al. [53] sorted out the opportunities and challenges faced by China's current development from three aspects: urban and rural digital divide, new consumption and new demand for safety and health. Therefore, this paper constructs a system of factors affecting PC's participation in university it, adopts the combination method of analytic hierarchy Process (AHP) and criterion importance through criterion correlation (AHP-Critic) to quantify the comprehensive assignment of influencing factors, which has certain advantages compared with existing relevant works.
We tried our best to improve the manuscript and made some changes marked in blue in revised paper which will not influence the content and framework of the paper. We appreciate for Editor/ Reviews’ warm work earnestly, and hope the correction will meet with approval. Once again, thank you very much for you comments and suggestions.
